# EXPERTSTEER: INTERVENING IN LLMS THROUGH EXPERT KNOWLEDGE

## ABSTRACT

Large Language Models (LLMs) exhibit remarkable capabilities across various tasks, yet guiding them to follow desired behaviours during inference remains a significant challenge. Activation steering offers a promising method to control the generation process of LLMs by modifying their internal activations. However, existing methods commonly intervene in the model's behaviour using steering vectors generated by the model itself, which constrains their effectiveness to that specific model and excludes the possibility of leveraging powerful *external expert models* for steering. To address these limitations, we propose **EXPERTSTEER**, a novel approach that leverages arbitrary specialized expert models to generate steering vectors, enabling intervention in any LLMs. EXPERTSTEER transfers the knowledge from an expert model to a target LLM through a cohesive four-step process: first aligning representation dimensions with auto-encoders to enable cross-model transfer, then identifying intervention layer pairs based on mutual information analysis, next generating steering vectors from the expert model using Recursive Feature Machines, and finally applying these vectors on the identified layers during inference to selectively guide the target LLM without updating model parameters. We conduct comprehensive experiments using three LLMs on 15 popular benchmarks across four distinct domains. Experiments demonstrate that EXPERTSTEER significantly outperforms established baselines at minimal cost.

## 1 INTRODUCTION

Large language models (LLMs) have demonstrated remarkable capabilities across diverse tasks (Anthropic, 2024; Reid et al., 2024; DeepSeek-AI et al., 2025; OpenAI, 2024a;b). However, aligning these LLMs with desirable behaviour remains challenging (Longpre et al., 2023; Ding et al., 2023; Wolf et al., 2024). Recent research attempts to address this challenge with prompt engineering (Brown et al., 2020; Wei et al., 2022c), supervised fine-tuning (SFT) (Wei et al., 2022b; Zhang et al., 2023; Bommasani et al., 2021), reinforcement learning from human feedback (RLHF) (Christiano et al., 2017; Ziegler et al., 2019; Bai et al., 2022a), which typically requires extensive resources. More recently, activation steering has been proposed as an alternative for these approaches. This technique modifies the LLMs' internal activations at inference time, which reduces the computational cost from fine-tuning and long context and prevents the catastrophic forgetting from updating the model parameters (Turner et al., 2023; Zou et al., 2023; Liu et al., 2022).

While activation steering has emerged as a promising approach, significant limitations hinder its broader applicability and effectiveness. Existing activation steering methods typically generate steering vectors using the model being steered itself (Li et al., 2023; Rimsky et al., 2024; Wang et al., 2024b; Liu et al., 2024). Consequently, these methods are constrained by the inherent knowledge of the LLM, which may lack the specialized expertise or deeper understanding required for certain tasks (Li et al., 2023; Rimsky et al., 2024; Wang et al., 2024c; Chen et al., 2024). Additionally, the steering vectors produced by these methods are limited to influencing the behaviour of the specific model they are derived from (Wang et al., 2024c; Cao et al., 2024; Bhattacharjee et al., 2024), making them unsuitable for cross-model steering and restricting their potential diverse applications. Moreover, as more powerful models with distinct strengths are developed, it becomes increasingly reasonable to consider leveraging these models as external resources for activation steering (Dong et al., 2023; You et al., 2022; Gu et al., 2024). Therefore, while activation steering holds significant promise as a flexible solution for effectively controlling LLM behaviours, its full potential remains underutilized.

To address these limitations, we introduce **EXPERTSTEER**, a novel activation steering framework that incorporates an arbitrary external expert model for generating steering vectors to effectively control the behaviours of any LLMs. To enable seamless cross-model steering, we first train auto-encoders (Hinton & Salakhutdinov, 2006) to align the hidden state dimensions of the expert model with those of the target LLM. Inspired by the Optimal Brain Surgeon principle (LeCun et al., 1989; Hassibi et al., 1993), we then perform mutual information analysis on the hidden states of both models to identify the optimal subset of layer pairs for intervention. Next, we extract informative features from the identified expert layers using Recursive Feature Machines (RFMs) Radhakrishnan et al. (2024), implemented through Kernel Ridge Regression (KRR) (Saunders et al., 1998) and Average Gradient Outer Product (AGOP) Radhakrishnan et al. (2024). The principal eigenvector of the resulting feature matrix for each identified expert layer is then used as the steering vector. Finally, the steering vectors are applied to the target LLM's hidden states at the identified intervention layers during inference time. By integrating auto-encoders, expert knowledge, and advanced feature extraction techniques, EXPERTSTEER provides an effective and efficient steering method that enables universal knowledge transfer between arbitrary pairs of models, making it a significant practical application.

To evaluate the effectiveness of EXPERTSTEER, we conduct extensive experiments involving three diverse LLMs and 15 widely recognized benchmarks spanning four domains: Medical, Financial, Mathematical, and General. Our study addresses two scenarios of knowledge transfer: from a domain-specific expert model to a general-purpose target LLM, and from a larger general-purpose model to a smaller general-purpose target LLM. The results show that EXPERTSTEER consistently outperforms previous steering methods across all tasks.

Our contributions are summarized as follows:

- We propose EXPERTSTEER, a novel activation steering approach that facilitates effective knowledge transfer from arbitrary expert models to any target LLMs. Leveraging techniques such as auto-encoders, mutual information analysis, and Recursive Feature Machines (RFMs), our method streamlines the steering process into four cohesive steps, extending the generalizability of activation steering and addressing the key limitations of existing approaches (see Section 3).
- We demonstrate the broad applicability and effectiveness of EXPERTSTEER across multiple models and tasks. Through extensive experiments with three LLMs over 15 diverse tasks spanning four domains, EXPERTSTEER consistently surpasses existing activation steering methods, underscoring the generalizability of EXPERTSTEER. (see Section 4).
- We provide a detailed analysis of EXPERTSTEER, focusing on the influence of feature extraction, expert selection, and the workflow of EXPERTSTEER. We also examine its computational efficiency, demonstrating EXPERTSTEER is highly cost-effective (see Section 5).

## 2 RELATED WORK

**Activation Steering** Activation steering provides a cost-effective way to steer model behaviours by directly manipulating activations during inference (Turner et al., 2023; Hernandez et al., 2023; Zou et al., 2023; Qiu et al., 2024). Current research based on steering vectors which are derived from activation differences in curated parallel positive-negative pairs enables interventions to change behaviours (Li et al., 2023; Chen et al., 2024; Liu et al., 2024; Cao et al., 2024; Bhattacharjee et al., 2024) or regulate the model's inference (Rimsky et al., 2024; Turner et al., 2023; Wang et al., 2024b; Stolfo et al., 2024) without the need for fine-tuning (Wei et al., 2022a; Bai et al., 2022b) or heavy in-context examples (Brown et al., 2020). However, current methods rely on the model itself to generate steering vectors, which restricts their effectiveness to the model's inherent knowledge and excludes the potential of utilizing external models (Rimsky et al., 2024; Tan et al., 2024).

**Knowledge Transfer** Knowledge transfer is a well-established techniques for performance improvement, where knowledge from a source model is transferred to a target model (Buciluǎ et al., 2006; Hinton et al., 2015). However, current methods, such as distillation via synthetic datasets (Kim & Rush, 2016; He et al., 2023; Hsieh et al., 2023; Zhou & Chiam, 2023) and teacher-student alignment (Jiao et al., 2020; Timiryasov & Tastet, 2023; Boizard et al., 2024), rely on computationally expensive fine-tuning and risk catastrophic forgetting. (Luo et al., 2023; Biderman et al., 2024). This underscores the need for more efficient, parameter-free knowledge transfer strategies.

Figure 1: An overview of EXPERTSTEER, including four steps: (1) aligning the dimensionality of the expert and target models, (2) identifying the layer pairs to be intervened upon, (3) generating steering vectors from the expert model, and (4) intervening in the generation process of the target model.

**Ours**   We propose EXPERTSTEER, a novel method that incorporates an arbitrary expert model for steering any LLMs, unlike prior approaches that generate steering vectors within the model itself (Li et al., 2023; Rimsky et al., 2024; Wang et al., 2024c). EXPERTSTEER effectively transfers the expertise to target LLMs via the steering vectors.

## 3   EXPERTSTEER

As illustrated in Figure 1, we elaborate each step of EXPERTSTEER in this section. The first step is to align the representations between the expert model and the target model, which is detailed in Section 3.1. Next, we identify the intervention layer pairs that exhibit significant differences in their representations, as described in Section 3.2. Following this, we generate steering vectors from the expert model using Recursive Feature Machines (RFMs) in Section 3.3. Finally, we apply these steering vectors to the target model during inference to enhance its performance, as outlined in Section 3.4. We provide implementation details in Section 3.5 and theoretical proofs in Appendix A.

### 3.1   REPRESENTATION ALIGNMENT

A significant challenge in transferring knowledge between different models is their architectural differences, particularly the varying dimensions of hidden states across models. To address this, we introduce a representation alignment procedure that unifies the feature spaces of the expert model and the target model. For each layer $i$ in the expert model with hidden states $h_i^E \in \mathbb{R}^{d_E}$, we train a dedicated auto-encoder consisting of an encoder $f_{\theta_i} : \mathbb{R}^{d_E} \to \mathbb{R}^{d_T}$ and a decoder $g_{\phi_i} : \mathbb{R}^{d_T} \to \mathbb{R}^{d_E}$, where $d_E$ and $d_T$ represent the hidden dimensions of the expert and target models, respectively. Here, both the encoder and decoder are implemented as one affine linear layer. The auto-encoder is optimized using a reconstruction loss function:

$$\mathcal{L}_{\text{recon}} = \frac{1}{K} \sum_{k=1}^{K} \|h_{i,k}^E - g_{\phi_i}(f_{\theta_i}(h_{i,k}^E))\|_2^2 \tag{1}$$

where $K$ is the number of training examples. This loss ensures that the encoder-decoder pair can effectively compress and expand the expert model's representations while preserving essential information. The trained encoder $f_{\theta_i}$ serves as a bridge between the expert and target feature spaces, enabling us to project the expert's hidden states into a form compatible with the target model.

### 3.2   INTERVENTION LAYER PAIRING

After aligning the representations between the expert and target models, the next step is to identify the layer-wise pairing relationship between the two models. Inspired by the Optimal Brain Surgeon (OBS) principle, which emphasizes that effective neural network modifications should be both selective and minimal (LeCun et al., 1989; Hassibi et al., 1993), we intervene in only a subset of the target model's layers to maximize the benefits of the intervention while minimizing the risk of introducing noise.

---

**Algorithm 1:** Recursive Feature Machines (RFMs)

**Input** : Training data $H_i = [h_{i,1}^E, h_{i,2}^E, \ldots, h_{i,K}^E] \in \mathbb{R}^{K \times d_E}$; binary labels
$Y = [y_1, y_2, \ldots, y_K] \in \mathbb{R}^K$; the number of iterations $\tau$; the bandwidth parameter $\sigma$;
the number of training examples $K$.

**Output :** Feature importance matrix $\mathcal{M}_i^\tau$

1  $\mathcal{M}_i^0 \leftarrow I_{d_E}$ ;                                    // Initialize feature importance matrix
2  **for** $t = 0$ *to* $\tau - 1$ **do**
3    $\mathbb{K}^t(h_{i,k}^E, z) \leftarrow \exp\left(-\frac{1}{\sigma}(h_{i,k}^E - z)^\top \mathcal{M}_i^t(h_{i,k}^E - z)\right)$ ;          // Update kernel function
4    $\beta_t \leftarrow \left(\mathbb{K}^t(H_i, H_i)\right)^{-1} Y$ ;                // Solve $\beta_t$ for the predictor $\pi^t(z) = \mathbb{K}^t(H_i, z)\beta_t$
5    $\mathcal{M}_i^{t+1} \leftarrow \frac{1}{K}\sum_{k=1}^{K} \nabla_{h_{i,k}^E}\pi^t(h_{i,k}^E) \cdot (\nabla_{h_{i,k}^E}\pi^t(h_{i,k}^E))^\top$ ;          // Compute AGOP matrix
6  **end**

---

Mutual information (MI) quantifies the amount of information obtained about one random variable through observing another random variable, making it an ideal metric for measuring representation alignment between two models. Hence, we conduct a layer-wise MI analysis to identify the layer pairs for steering. For each layer pair $(i, j)$, where $i$ refers to a layer in the expert model and $j$ refers to a layer in the target model, we follow Zheleznlak et al. (2020) to estimate the MI as follows:

$$\text{MI}(i, j) = \frac{1}{K}\sum_{k=1}^{K}\mathbb{I}(f_{\theta_i}(h_{i,k}^E); h_{j,k}^T), \quad \text{where} \quad \mathbb{I}(X; Y) = \int\int p(x, y)\log\frac{p(x, y)}{p(x)p(y)}dxdy \quad (2)$$

Here, $\mathbb{I}(\cdot; \cdot)$ denotes the mutual information operator, measuring the reduction in uncertainty about $Y$ when $X$ is known, and $K$ is the number of examples used for estimate MI. For $k$-th example, the expert's hidden states at $i$-th layer $h_{i,k}^E$ are mapped to the target's dimensionality by the encoder $f_{\theta_i}$, and $h_{j,k}^T$ represents the hidden states of the target model at layer $j$. Lower MI indicates a greater disparity between the expert layer and the target layer, implying that the representation at the target layer potentially lacks the expert's knowledge. This suggests a greater need for intervention. Conversely, higher MI implies that the target model's representation is already well-aligned with the expert's, thereby reducing the necessity for intervention.

Subsequently, we select intervention points where knowledge transfer would be most beneficial. Specifically, we compute the MI for all layer pairs $(i, j)$ and select the top-$P$ pairs with the lowest values. These low-MI pairs represent areas where the target's representations diverge most significantly from the expert's knowledge, making them better candidates for intervention.

### 3.3 STEERING VECTOR GENERATION

After identifying the intervention layer pairs, we need to generate steering vectors that encode the expert model's knowledge. To this end, we employ Recursive Feature Machines (RFMs) Radhakrishnan et al. (2024) to extract the most informative features from the expert model's hidden states. In our approach, the RFMs algorithm employs two key components: Kernel Ridge Regression (KRR) (Saunders et al., 1998) and the Adaptive Gradient Optimal Perturbation (AGOP) matrix Radhakrishnan et al. (2024). The KRR model learns to distinguish between hidden states given by inputs from different sources by binary classification, while the AGOP matrix captures the feature importance by analysing gradients of the KRR model.

For each selected expert model layer $i$, we gather hidden states $H_i = [h_{i,1}^E, h_{i,2}^E, \ldots, h_{i,K}^E] \in \mathbb{R}^{K \times d_E}$ from $K$ training examples. Each example is assigned a binary label with One-vs-Rest strategy: positive (1) for examples that align with the expert's knowledge, and negative (0) for examples that do not. For instance, when using a medical LLM as the expert, examples related to medical topics are labelled as positive, while examples unrelated to the medical domain are labelled as negative.

As detailed in Algorithm 1, in each iteration $t$ of the RFMs, we first update the Mahalanobis Laplace Kernel function $\mathbb{K}^t$ using the current feature importance matrix $\mathcal{M}_i^t$ (line 3), where $z$ in $\mathbb{K}_i^t$ indicates an arbitrary hidden state from $H_i$. This adaptive kernel measures the similarity between hidden states while accounting for their relevance to domain distinction. We then solve for coefficients

$\beta_t$ using KRR, which optimizes the predictor $\pi^t(z) = \mathbb{K}^t(H_i, z)\beta_t$ to classify representations by domain (line 4). Finally, we update the feature importance matrix $\mathcal{M}_i^{t+1}$ by computing AGOP, which averages the outer products of gradients across all training examples (line 5). After $\tau$ iterations, the final matrix $\mathcal{M}_i^\tau$ captures directions in the feature space that most reflect desired knowledge.

To extract the steering vector from this feature importance matrix, we perform eigenvalue decomposition on $\mathcal{M}_i^\tau = U \Lambda U^\top$, where $\Lambda = \text{diag}(\lambda_1, \lambda_2, \ldots, \lambda_{d_E})$ are the eigenvalues (sorted in descending order) and $U = [u_1, u_2, \ldots, u_{d_E}]$ are the corresponding eigenvectors. The eigenvector $u_1$ associated with the largest eigenvalue $\lambda_1$ represents the direction of maximum variance in the feature space, capturing the most desired knowledge. We define $u_1$ as the steering vector $\nu_i$ for the $i$-th layer. This approach ensures that our intervention targets the most salient aspects of the expert model's knowledge, maximizing the effectiveness of the knowledge transfer.

### 3.4 EXPERTISE INTERVENTION

In the final step, we transfer the expert knowledge distilled in the steering vectors to the target model by intervening at the $P$ most impactful layer pairs $(i, j)$ identified previously. Since the expert and target models may have different hidden dimensions ($d_E$ and $d_T$), we ensure compatibility by leveraging the encoder $f_{\theta_i}(\cdot)$ from the trained auto-encoder (see Section 3.1). This encoder projects the expert's steering vector $\nu_i \in \mathbb{R}^{d_E}$ into the target model's feature space $\mathbb{R}^{d_T}$ when necessary. Formally, for each selected layer pair $(i, j)$, we update the hidden state $h_j^T$ of the target model:

$$\hat{h}_j^T = \begin{cases} h_j^T + \varepsilon \cdot f_{\theta_i}(\nu_i) & \text{if } d_E \neq d_T \\ h_j^T + \varepsilon \cdot \nu_i & \text{if } d_E = d_T \end{cases} \tag{3}$$

where $\varepsilon$ is a scaling factor controlling the strength of the intervention. The modified hidden state $\hat{h}_j^T$ is then propagated through the remaining layers of the target model to produce the final output.

### 3.5 IMPLEMENTATION DETAILS

Our method introduces two hyperparameters: $P \in \mathbb{N}^+$, specifying the number of top layer pairs selected for intervention, and $\varepsilon \in \mathbb{R}^+$, controlling the strength of the intervention. In our experiments, we explore $P$ values ranging from 1 to 10, and $\varepsilon$ values in $\{1, 2, 4, 6, 8, 10, 12, 14, 16\}$. Following (Li et al., 2023; Rimsky et al., 2024; Wang et al., 2024c), we perform a hyperparameter sweep to empirically determine the optimal settings on a small development set, which are subsequently utilized during the final evaluation on the test set.

We use 2,000 random examples to train the auto-encoders in Section 3.1. Then, we leverage 500 random examples to identify the intervention pairs in Section 3.2. And, we sample 2,000 positive examples and 2,000 negative examples to train RFMs in Section 3.3. More details are in Appendix D.

## 4 EXPERIMENTS

### 4.1 EXPERIMENTAL SETUP

**Datasets** We conduct our experiments across four domains: Medical, Financial, Mathematical, General and present the datasets in Table 1. We denote the overall performance within one domain as $\mu_{\text{ALL}}$, which is the macro-average of the tasks. More details are provided in Appendix B.

Table 1: The datasets used for training and evaluation, and the expert models utilized in this work.

| | Training Datasets | Evaluation Datasets | | | Expert Model |
|---|---|---|---|---|---|
| Medical | UltraMedical | MedQA, MedMCQA, MMLU-Medical | | | Bio-Medical-Llama-3-8B |
| Financial | FINQA | FPB, Flare-cfa, MMLU-Financial | | | Llama-3-8B-Instruct-Finance |
| Mathematical | MetaMathQA | GSM8K, MATH500, MMLU-Math | | | Qwen2.5-Math-7B-Instruct |
| General | LMSYS-Chat-1M | COPA, MMLU-Humanities, Harmful Behaviors | NLI, | ARC-C, Salad, | Qwen2.5-14B-Instruct |

Table 2: Results on the Medical, Financial and Mathematical domains with Llama-3.1-8B-Instruct, Qwen2.5-7B-Instruct, and Gemma-2-2b-Instruct target models across discriminative tasks and generative tasks . The expert models are Bio-Medical-Llama-3-8B, Llama-3-8B-Instruct-Finance and Qwen2.5-Math-7B-Instruct. Same-Family ($\mathcal{SF}$) and Cross-Family ($\mathcal{XF}$) indicates that if the expert and target model belong to the same model family. The best overall results are highlighted.

| | Medical | | | | Financial | | | | Mathematical | | | |
|---|---|---|---|---|---|---|---|---|---|---|---|---|
| | $\mu_{\text{ALL}}$ | MedQA | Med MCQA | MMLU Med. | $\mu_{\text{ALL}}$ | FPB | Flare -cfa | MMLU Fin. | $\mu_{\text{ALL}}$ | MMLU Math | GSM8K | MATH 500 |
| Expert Model | 76.61 | 73.85 | 69.01 | 86.96 | 60.01 | 64.34 | 59.49 | 56.20 | 58.55 | 25.09 | 91.60 | 58.95 |
| **Llama-3.1-8B-Instruct** | | | | | | | | | | | | |
| Baseline | 52.00 | 45.60 | 49.40 | 60.99 | 45.98 | 41.55 | 48.14 | 48.26 | 51.43 | 25.48 | 86.80 | 42.00 |
| SFT | 56.44 | 53.50 | 51.35 | 64.46 | 55.73 | 54.84 | 56.00 | 56.35 | 46.17 | 22.51 | 80.00 | 36.00 |
| KD | 56.06 | 53.56 | 48.98 | 65.65 | 56.16 | 55.11 | 56.68 | 56.70 | 44.91 | 21.32 | 78.80 | 34.60 |
| ITI | 54.34 | 50.71 | 50.11 | 62.20 | 49.01 | 47.80 | 49.91 | 49.31 | 52.86 | 29.17 | 87.00 | 42.40 |
| CAA | 46.60 | 38.86 | 45.72 | 55.22 | 47.39 | 50.21 | 46.23 | 45.72 | 34.83 | 26.10 | 55.20 | 23.20 |
| SADI | 53.51 | 50.51 | 47.02 | 62.99 | 49.61 | 51.96 | 47.00 | 49.87 | 52.62 | 28.07 | 87.00 | 42.80 |
| SAS | 52.33 | 46.20 | 49.40 | 61.40 | 47.63 | 47.45 | 45.56 | 49.87 | 52.12 | 26.75 | 86.80 | 42.80 |
| EXPERTSTEER | 56.98 | 53.59 | 50.66 | 66.71 | 51.49 | 55.21 | 48.92 | 50.35 | 54.92 | 31.95 | 88.40 | 44.40 |
| | | $\mathcal{SF}$ | | | | $\mathcal{SF}$ | | | | $\mathcal{XF}$ | | |
| **Qwen2.5-7B-Instruct** | | | | | | | | | | | | |
| Baseline | 49.65 | 41.20 | 46.25 | 61.50 | 65.53 | 76.23 | 57.88 | 62.49 | 55.05 | 26.75 | 89.20 | 49.20 |
| SFT | 55.55 | 45.30 | 51.02 | 70.32 | 67.73 | 74.59 | 59.78 | 68.83 | 53.48 | 30.04 | 83.20 | 47.20 |
| KD | 53.20 | 43.68 | 47.68 | 68.23 | 66.44 | 76.59 | 58.36 | 64.37 | 56.88 | 31.03 | 90.80 | 48.80 |
| ITI | 49.55 | 41.46 | 45.78 | 61.40 | 60.25 | 76.42 | 42.47 | 61.86 | 49.85 | 11.14 | 90.00 | 48.40 |
| CAA | 50.04 | 41.46 | 46.18 | 62.48 | 65.65 | 76.63 | 56.54 | 63.79 | 42.48 | 11.85 | 81.20 | 34.40 |
| SADI | 50.38 | 42.34 | 45.72 | 63.09 | 66.24 | 76.91 | 57.95 | 63.88 | 52.95 | 22.05 | 88.80 | 48.00 |
| SAS | 49.67 | 41.20 | 46.42 | 61.40 | 65.84 | 76.59 | 56.68 | 64.26 | 52.09 | 19.08 | 88.80 | 48.40 |
| EXPERTSTEER | 54.03 | 45.98 | 48.57 | 67.53 | 70.87 | 78.40 | 63.23 | 71.00 | 57.26 | 31.17 | 90.80 | 49.80 |
| | | $\mathcal{XF}$ | | | | $\mathcal{XF}$ | | | | $\mathcal{SF}$ | | |
| **Gemma-2-2b-Instruct** | | | | | | | | | | | | |
| Baseline | 31.17 | 28.63 | 33.06 | 31.81 | 37.15 | 47.27 | 36.00 | 28.17 | 37.94 | 23.03 | 67.60 | 23.20 |
| SFT | 40.60 | 37.13 | 32.74 | 51.93 | 48.76 | 53.29 | 46.67 | 46.33 | 35.11 | 23.33 | 57.60 | 24.40 |
| KD | 39.79 | 35.80 | 33.19 | 50.39 | 46.54 | 50.71 | 45.56 | 43.33 | 33.99 | 21.17 | 56.80 | 24.00 |
| ITI | 31.23 | 28.78 | 33.12 | 31.80 | 37.61 | 48.25 | 36.09 | 28.50 | 36.75 | 21.46 | 68.00 | 20.80 |
| CAA | 30.65 | 28.17 | 32.65 | 31.14 | 37.15 | 46.85 | 36.59 | 28.02 | 35.75 | 22.85 | 61.20 | 23.20 |
| SADI | 30.99 | 28.99 | 32.03 | 31.96 | 38.41 | 49.90 | 36.63 | 28.71 | 37.62 | 22.05 | 67.60 | 23.20 |
| SAS | 30.61 | 27.46 | 32.74 | 31.62 | 37.32 | 47.80 | 35.67 | 28.50 | 35.97 | 21.32 | 61.80 | 24.80 |
| EXPERTSTEER | 32.21 | 29.39 | 33.37 | 33.87 | 39.47 | 51.21 | 37.40 | 29.80 | 39.28 | 24.24 | 68.40 | 25.20 |
| | | $\mathcal{XF}$ | | | | $\mathcal{XF}$ | | | | $\mathcal{XF}$ | | |

**Model Backbones** We apply EXPERTSTEER to three diverse target models from different families and sizes: Llama-3.1-8B-Instruct (Dubey et al., 2024), Qwen2.5-7B-Instruct (Yang et al., 2024), and Gemma-2-2b-Instruct (Rivière et al., 2024). The expert models are show in Table 1.

**Baselines** We compare EXPERTSTEER with predictions give by the original LLMs (Baseline), several fine-tuning baselines: standard Supervised Fine-Tuning (SFT) and Knowledge Distillation (KD) (Boizard et al., 2024), and the state-of-the-art steering baselines, including Inference-Time Intervention (ITI) (Li et al., 2023),Contrastive Activation Addition (CAA) (Rimsky et al., 2024), Semantic-Adaptive Dynamic Intervention (SADI) (Wang et al., 2024c), and Sparse Activation Steering (SAS) (Bayat et al., 2025). More details are shown in Appendix C.

## 4.2 OVERALL PERFORMANCE

**EXPERTSTEER effectively transfers domain-specific knowledge and significantly enhances model performance on both discriminative and generative tasks.** As shown in Table 2, EXPERT-

STEER consistently boosts performance across three target models and three domains, outperforming other intervention methods and matching or surpassing fully fine-tuned approaches like SFT and KD. In the Medical and Financial domains, it provides average gains of +4.98 for Llama-3.1-8B-Instruct and +5.34 for Qwen2.5-7B-Instruct. Furthermore, EXPERTSTEER consistently outperforms SFT and KD in the Mathematical domain, demonstrating its superior efficiency for highly complex tasks. Even when target models occasionally outperforms expert models, EXPERTSTEER discovers additional knowledge through steering vectors. For example, on the FPB benchmark, the Qwen2.5-7B-Instruct baseline and expert models achieve scores of 76.23 and 64.34, respectively, while EXPERTSTEER achieves 78.40. This underscores the effectiveness of EXPERTSTEER in transferring expertise.

**EXPERTSTEER consistently excels in both same-family and cross-family settings.** In practice, expert and target models are likely to come from different families. Hence, we evaluate EXPERT-STEER under both same-family ($\mathcal{SF}$) and cross-family ($\mathcal{XF}$) settings, where same-family indicates that the expert model and the target model belong to the same model family, while cross-family indicates that they belong to different families. As shown in Table 2, EXPERTSTEER consistently outperforms the baseline in both settings, showing gains of +4.98, +5.51, and +1.34 in three domains for same-family, and +4.38 (Medical) and +5.34 (Financial) in cross-family settings using Qwen2.5-7B-Instruct as the target. These results confirm that EXPERTSTEER effectively extracts and transfers expertise despite model disparities, demonstrating its applicability and generalizability.

**EXPERTSTEER can also improve the model performance when the expert and target models share the same domain.** While EXPERTSTEER effectively transfers knowledge across domains, we also investigate its potential to enhance model performance when both the expert and target models belong to the same domain. To this end, we leverage the general-purpose Qwen2.5-14B-Instruct as the expert model and present the results in Table 3. The results demonstrate that EXPERTSTEER consistently outperforms other steering methods on both natural language understanding (NLU) and safety tasks. Unlike prior steering methods, which are often constrained by the model's inherent capabilities, EXPERTSTEER effectively leverages the strengths of more powerful models, thereby unlocking their full potential. These findings highlight the versatility and effectiveness of EXPERTSTEER in both cross-domain and same-domain scenarios.

Table 3: General domain performance on the NLU tasks and Safety tasks across discriminative tasks and generative tasks . The expert model is Qwen2.5-14B-Instruct.

| | NLU | | | | | Safety | | |
|---|---|---|---|---|---|---|---|---|
| | $\mu_{ALL}$ | COPA | NLI | ARC-C | MMLU Hum. | $\mu_{ALL}$ | Salad | Harm Behav. |
| Expert Model | 82.42 | 96.60 | 75.66 | 82.72 | 74.71 | 83.20 | 78.40 | 88.00 |
| **Llama-3.1-8B-Instruct** $\mathcal{XF}$ | | | | | | | | |
| Baseline | 64.68 | 74.01 | 57.87 | 67.39 | 59.45 | 65.20 | 57.20 | 73.20 |
| ITI | 67.01 | 81.75 | 57.82 | **68.97** | 59.49 | 72.60 | **72.00** | 73.20 |
| CAA | 63.11 | 80.90 | 50.47 | 64.13 | 56.93 | 72.40 | 71.60 | 73.20 |
| SADI | 65.32 | 81.36 | 57.98 | 64.93 | 57.00 | 72.20 | 71.60 | 72.80 |
| SAS | 64.64 | 80.60 | 56.42 | 64.13 | 57.41 | 72.60 | **72.00** | 73.20 |
| EXPERTSTEER | **68.45** | **83.47** | **61.36** | 68.34 | **60.60** | **72.80** | **72.00** | **73.60** |
| **Qwen2.5-7B-Instruct** $\mathcal{SF}$ | | | | | | | | |
| Baseline | 72.51 | 82.07 | 71.00 | 73.54 | 63.41 | 77.60 | 72.80 | 82.40 |
| ITI | 72.74 | 82.03 | 73.63 | 72.95 | 62.34 | 80.20 | 75.60 | 84.80 |
| CAA | 73.66 | 84.20 | 73.25 | 74.26 | 62.94 | **82.40** | 74.80 | **90.00** |
| SADI | 74.08 | 85.37 | 73.99 | 73.98 | 62.98 | 81.30 | **76.00** | 86.60 |
| SAS | 72.69 | 81.75 | 72.87 | 73.54 | 62.60 | 80.00 | 74.80 | 85.20 |
| EXPERTSTEER | **77.53** | **88.23** | **77.20** | **78.24** | **66.44** | 79.20 | 74.80 | 83.60 |
| **Gemma-2-2b-Instruct** $\mathcal{XF}$ | | | | | | | | |
| Baseline | 46.67 | 72.32 | 41.82 | 38.17 | 34.36 | 78.60 | 74.80 | 82.40 |
| ITI | 46.38 | 73.34 | 40.21 | 37.85 | 34.11 | 80.80 | 77.60 | 84.00 |
| CAA | 45.73 | 69.75 | 42.38 | 37.40 | 33.39 | **81.30** | 78.00 | **84.60** |
| SADI | 45.76 | 71.14 | 40.50 | 37.03 | 34.36 | 81.10 | 78.00 | 84.20 |
| SAS | 45.95 | 70.86 | 41.47 | 37.03 | 34.42 | 80.20 | 76.80 | 83.60 |
| EXPERTSTEER | **48.35** | **75.57** | **44.10** | **39.11** | **34.63** | **81.30** | **78.40** | 84.20 |

**EXPERTSTEER effectively transfers linguistic expertise.** While our primary experiments focus on English, we extend EXPERTSTEER to other languages to demonstrate its broader applicability. Specifically, we evaluate EXPERTSTEER on Chinese datasets: XCOPA-zh (Ponti et al., 2020), XNLI-zh (Conneau et al., 2018), XStoryCloze-zh (Lin et al., 2021), Flores-en2zh, and

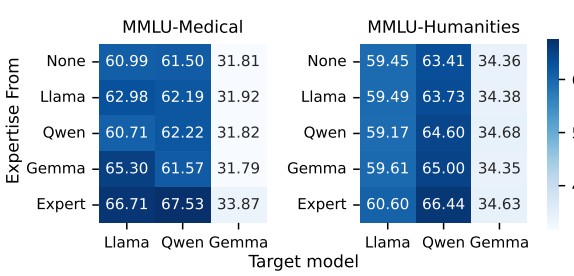
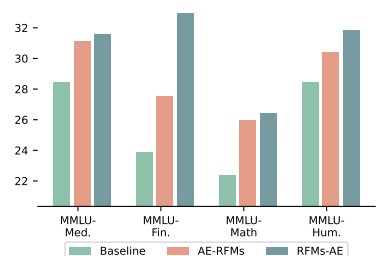

Figure 2: The selection of model for generating steering vectors. "None" indicates no expert is used. "Expert" represents the models in Table 1. "Llama", "Qwen", "Gemma" represent Llama-3.1-8B-Instruct, Qwen2.5-7B-Instruct, and Gemma-2-2b-Instruct, respectively.

Figure 3: Comparison of *RFMs-AE* and *AE-RFMs* using Llama-3.2-1B-Instruct: *RFMs-AE* extracts features before aligning dimensions, yet *AE-RFMs* aligns dimensions before feature extraction.

`Flores-zh2en` (Costa-Jussà et al., 2022), using the expert model Llama3.1-8B-Chinese-Chat (Wang et al., 2024a). The steering vector is extracted from 2,000 items randomly selected from the Chinese News Commentary dataset. Results in Table 4 show consistent performance gains for both Llama-3.1-8B-Instruct and Qwen2.5-7B-Instruct, confirming that the effectiveness of EXPERTSTEER extends beyond English.

Table 4: Chinese results with Llama3.1-8B-Chinese-Chat as expert model. `xsc` represents XStoryCloze.

| $\mu_{\text{ALL}}$ | XCOPA -zh | XNLI -zh | xsc -zh | Flores -en2zh | Flores -zh2en |
|---|---|---|---|---|---|
| Expert Model | 57.58 | 87.13 | 60.14 | 87.86 | 32.79 | 19.96 |
| Llama-3.1-8B-Instruct | | | | | | |
| Baseline | 49.56 | 77.58 | 49.17 | 76.10 | 26.36 | 18.58 |
| EXPERTSTEER | 50.98 | 78.32 | 49.63 | 76.39 | 31.11 | 19.46 |
| Qwen2.5-7B-Instruct | | | | | | |
| Baseline | 58.22 | 79.60 | 63.39 | 93.25 | 34.95 | 19.90 |
| EXPERTSTEER | 62.82 | 91.69 | 71.90 | 94.85 | 35.05 | 20.62 |

## 5 ANALYSIS

In this section, we firstly conduct ablation studies to analyse EXPERTSTEER in Section 5.1, including the impact of feature extraction methods, the choice of the expert models, and the order of operations. We examine the computational efficiency and explore how the foundation models affect performance of EXPERTSTEER in Section 5.2. We present more analyses in Appendix E.

### 5.1 ABLATION STUDIES

**RFMs excel in feature extraction.** Unlike linear activation steering methods, EXPERTSTEER uses RFMs with a non-linear kernel to extract steering vectors. To validate effectiveness of RFMs, we compare RFMs with linear approaches, such as mean difference (MD) and Principal Component Analysis (PCA) on the medical and general tasks. As shown in Table 5, EXPERTSTEER with RFMs consistently outperforms those with MD or PCA across all evaluations.

Table 5: Comparison between different feature extraction methods.

| | MedQA | MMLU Med. | COPA | MMLU Hum. |
|---|---|---|---|---|
| Llama-3.1-8B-Instruct | | | | |
| Baseline | 45.60 | 60.99 | 74.01 | 59.45 |
| EXPERTSTEER | | | | |
| ├ MD | 42.56 | 59.11 | 81.19 | 59.88 |
| ├ PCA | 42.58 | 59.17 | 83.10 | 59.89 |
| └ RFMs | **53.59** | **66.71** | **83.47** | **60.60** |
| Qwen2.5-7B-Instruct | | | | |
| Baseline | 41.20 | 61.50 | 82.07 | 63.41 |
| EXPERTSTEER | | | | |
| ├ MD | 43.18 | 64.97 | 82.01 | 64.10 |
| ├ PCA | 44.52 | 65.30 | 86.01 | 64.59 |
| └ RFMs | **45.98** | **67.53** | **88.23** | **66.44** |

Among linear methods, PCA often exceeds MD by capturing higher-dimensional variance, while MD only considers first-order statistical differences between domains. More results are in Appendix H.

**The choice of the expert model is essential for activation steering.** Expert model selection is vital for EXPERTSTEER. As illustrated in Figure 2, we evaluate the performance of EXPERTSTEER using steering vectors generated by various models, including general-purpose models (Llama, Qwen, and Gemma) and expert models. We observe that steering vectors from experts significantly outperform those from general-purpose models, as they better capture most salient desired features. For instance, applying Llama-3.1-8B-Instruct on itself yields only a slight improvement (62.98 versus baseline 60.99 on `MMLU-Medical`), whereas expert models deliver a substantial boost (e.g., 66.23).

Furthermore, we observe similar patterns on the MMLU-Humanities in Figure 2. These findings highlight the limitations of the model itself, which relies on its inherent knowledge, whereas expert models are better equipped to generate effective steering vectors. More results are in Appendix I.

**It is essential for EXPERTSTEER to first extract features and subsequently align the representations.** As shown in Figure 1, we first extract hidden-state features from the expert model, align them to the target models with trained auto-encoders, and then perform the intervention. We refer to this approach as *RFMs-AE*. Alternatively, we can first align the sizes of hidden states using auto-encoders and then extract steering vectors by modifying Algorithm 1 line 4 as follows:

$$\pi^t(z) = \mathbb{K}^t(f_{\theta_i}(H_i), f_{\theta_i}(z))\beta_t, \quad \text{where} \quad \beta_t = \left(\mathbb{K}^t(f_{\theta_i}(H_i), f_{\theta_i}(H_i))\right)^{-1} Y \tag{4}$$

This approach is referred to as *AE-RFMs*. Experimental results in Figure 3 show that *RFMs-AE* consistently outperforms *AE-RFMs*. This indicates that applying RFMs directly to raw hidden states preserves the integrity of the original feature space during the critical feature extraction phase, capturing nuanced patterns that might otherwise be lost with dimensionality reduction. This aligns with multimodal fusion research, which indicates that feature extraction prior to dimensionality reduction enhances performance (Baltrusaitis et al., 2019). By retaining original features during extraction, our approach generates more informative steering vectors for intervention.

## 5.2 DISCUSSION

**EXPERTSTEER demonstrates high computational efficiency.** As shown in Figure 4, increasing the training data volume linearly increases training time without necessarily improving accuracy on MMLU-Medical and MMLU-Humanities tasks. We demonstrate that 2,000 training examples are sufficient for generating effective steering vectors, with an affordable time cost of approximately 17 minutes. Moreover, as detailed in Equation 3, EXPERTSTEER adds only a single constant vector per layer. By adding $\varepsilon \cdot f_{\theta_i}(\nu_i)$ to the hidden states as a bias term, our intervention imposes negligible computational overhead during inference, highlighting the efficiency of our method, making it both scalable and practical.

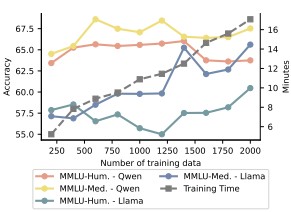

Figure 4: Training cost with different target models.

**EXPERTSTEER demonstrates effectiveness when using base models as the target models.** Building on our earlier findings that EXPERTSTEER boosts performance, we now explore its impact on base models by applying it to MMLU tasks with Llama-3.1-8B and Qwen2.5-7B. As shown in Table 6, although EXPERTSTEER again improves results, the gains are smaller than with SFT target models, referring to Table 2, because the steering vectors (derived from SFT expert models) face a larger distributional gap when applied to base models. This gap reduces effectiveness of the steering vectors in transferring expertise.

Table 6: Results of EXPERTSTEER using base model as the target model.

| | MMLU Med. | MMLU Fin. | MMLU Hum. | MMLU Math |
|---|---|---|---|---|
| | Llama-3.1-8B | | | |
| Baseline | 25.11 | 26.64 | 25.22 | 24.33 |
| EXPERTSTEER | **26.29** | **30.69** | **26.56** | **26.39** |
| | Qwen2.5-7B | | | |
| Baseline | 57.92 | 59.87 | 59.17 | 36.70 |
| EXPERTSTEER | **59.14** | **60.97** | **59.55** | **38.28** |

## 6 CONCLUSION

In this work, we introduce EXPERTSTEER, a novel activation steering method designed to enable knowledge transfer from any expert model to arbitrary target LLMs. Our approach consists of four key steps: (1) aligning the dimensionalities of the expert and target models using auto-encoders, (2) identifying optimal layer pairs for intervention through mutual information analysis, (3) generating steering vectors via Recursive Feature Machines (RFMs) from the identified expert layers, and (4) applying these steering vectors to the identified target layers. Extensive experiments across diverse LLMs and various tasks demonstrate that EXPERTSTEER significantly improve the performance, confirming its effectiveness and generalizability. This study advances the activation steering research in LLMs by introducing an effective and efficient intervention technique.

ETHICS STATEMENT

This work introduces EXPERTSTEER, a method for intervening in large language models (LLMs) through expert knowledge transfer using activation steering. All experiments were conducted using publicly available models and datasets, strictly adhering to their respective licenses and terms of use. No human subjects were involved in this research. We have conducted extensive experiments across diverse tasks to evaluate the effectiveness of EXPERTSTEER, but we recognize that biases in the underlying models and datasets may still affect outcomes. We encourage responsible use of EXPERTSTEER, with attention to fairness, transparency, and accountability in deployment.

REPRODUCIBILITY STATEMENT

We are committed to the reproducibility of our findings. Comprehensive details of the EXPERTSTEER methodology, including representation alignment, intervention layer selection, steering vector generation, and expertise intervention, are provided in Section 3. Experimental setups, including model configurations, datasets, and evaluation metrics, are described in Section 4 and Appendix D. All models and datasets used are publicly available, with references and links included for accessibility. To further support reproducibility, we will release our code and scripts upon publication, enabling other researchers to replicate and extend our results.

THE USE OF LARGE LANGUAGE MODELS (LLMS)

In preparing this work, we utilize large language models (LLMs) as general-purpose tools to assist with writing polish and grammar correction. The LLMs are not involved in research ideation, experimental design, or substantive content generation. Their role is limited to improving the clarity and readability of the text, ensuring grammatical accuracy, and refining the presentation of our findings. All scientific contributions, analyses, and conclusions are solely the work of the authors.

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

# A  THEORETICAL PROOFS

## A.1  MUTUAL INFORMATION AND REPRESENTATION DIVERGENCE

Let $X$ be the input, and let $E_l^A(X)$ and $E_l^B(X)$ denote the representations at layer $l$ of expert model $A$ and target model $B$, respectively. The mutual information (MI) between $E_l^A(X)$ and $E_l^B(X)$ is defined as:

$$I(E_l^A(X); E_l^B(X)) = \int p(e^A, e^B) \log \frac{p(e^A, e^B)}{p(e^A)p(e^B)} \, de^A de^B$$

Low MI implies that the representations encode largely independent information, indicating that the feature spaces are misaligned. Here, the input $X$ is drawn from a specific domain (e.g., medical), where the expert model typically demonstrates superior performance compared to the target model. Low MI suggests that the two models capture different domain-specific knowledge, which leads to divergent performance and highlights the potential for knowledge transfer. Conversely, high MI indicates that the layer pairs encode similar information, making alignment unnecessary.

## A.2  THEORETICAL ANALYSIS OF RFMs

In this appendix we provide a proof sketch establishing the convergence and fixed-point properties of Recursive Feature Machines (RFMs). The analysis formalizes why recursive updates yield steering vectors that capture expert knowledge.

**Setup and Notation**  Let $H = \{h_k\}_{k=1}^K \subset \mathbb{R}^d$ denote hidden states with labels $y_k \in \{0, 1\}$. At iteration $t$, RFMs maintain a positive semidefinite (PSD) matrix $\mathcal{M}^t \in \mathbb{R}^{d \times d}$ and define the Mahalanobis Laplace kernel:

$$\mathbb{K}^t(x, z) = \exp\left(-\tfrac{1}{\sigma}(x - z)^\top \mathcal{M}^t (x - z)\right). \tag{5}$$

Kernel ridge regression (KRR) with regularization $\lambda > 0$ yields predictor

$$\pi^t(z) = \mathbb{K}^t(H, z) \, \beta_t, \quad \beta_t = \left(\mathbb{K}^t(H, H) + \lambda I\right)^{-1} Y, \tag{6}$$

where $Y = (y_1, \ldots, y_K)^\top$. The average gradient outer product (AGOP) is defined as

$$\mathcal{G}(\mathcal{M}^t) = \frac{1}{K} \sum_{k=1}^K \nabla_z \pi^t(h_k) \, \nabla_z \pi^t(h_k)^\top. \tag{7}$$

RFMs update $\mathcal{M}^{t+1} \leftarrow \mathcal{G}(\mathcal{M}^t)$, optionally normalized to enforce $\mathrm{tr}(\mathcal{M}^{t+1}) = 1$.

**Assumptions**

- (A1) **Bounded domain:** Hidden states lie in a compact set, so all kernel values and gradients are uniformly bounded.
- (A2) **Regularization:** $\lambda > 0$ ensures the KRR solution $\beta_t$ is stable and continuous in $\mathcal{M}^t$.
- (A3) **Normalization:** Each update $\mathcal{M}^{t+1}$ is scaled to lie in the compact convex set $\mathbb{M} = \{\mathcal{M} \succeq 0 : \mathrm{tr}(\mathcal{M}) = 1\}$.

**Fixed-Point Property**

**Theorem 1** *Under assumptions (A1)–(A3), the map $\Phi : \mathcal{M} \mapsto \mathcal{G}(\mathcal{M})$ has at least one fixed point $\mathcal{M}^* \in \mathbb{M}$ such that*

$$\mathcal{M}^* \propto \mathcal{G}(\mathcal{M}^*). \tag{8}$$

Continuity of the kernel map $\mathcal{M} \mapsto \mathbb{K}_{\mathcal{M}}(H, H)$ follows from boundedness and smooth dependence of the Laplace kernel on $\mathcal{M}$. Since $\beta_t$ is a continuous function of $\mathbb{K}_{\mathcal{M}}(H, H)$ (due to $\lambda > 0$), and gradients of $\pi_t$ are linear in $\beta_t$, the operator $\Phi(\mathcal{M})$ is continuous. Because $\mathcal{M}$ is compact and convex, Brouwer's fixed-point theorem guarantees the existence of $\mathcal{M}^*$.

**Convergence Intuition**   Linearizing $\Phi$ near a fixed point $\mathcal{M}^*$ shows that $\mathcal{G}(\mathcal{M})$ acts as a covariance operator over predictor gradients. Iteration $\mathcal{M}^{t+1} = \Phi(\mathcal{M}^t)$ is therefore analogous to the power method on this covariance operator: the dominant eigenspace of $\mathcal{G}(\mathcal{M})$ is amplified across iterations. With trace normalization, $\mathcal{M}^t$ converges to a matrix whose top eigenvectors span the task-relevant subspace.

**Implications**   The fixed-point analysis explains why RFMs extract effective steering vectors: the top eigenvector of $\mathcal{M}^*$ represents the direction in hidden space where predictor gradients exhibit maximal variance, i.e., the direction most influential for the task. This provides a theoretical foundation for the empirical finding that RFMs with Laplace kernels outperform linear feature extraction methods such as MD and PCA.

### A.3   THE CHOICE OF LAPLACE KERNELS

While RFMs can be instantiated with any positive definite kernel, we employ the Mahalanobis Laplace kernel,

$$\mathbb{K}_{\mathcal{M}}(x, z) = \exp\Big( - \tfrac{1}{\sigma} (x - z)^\top \mathcal{M} (x - z) \Big), \tag{9}$$

where $\mathcal{M} \succeq 0$ is the adaptive metric and $\sigma > 0$ a bandwidth parameter. Below we motivate this choice.

**Gradient behaviour.**   The steering mechanism in RFMs depends on the gradients of the kernel predictor. For a general Mahalanobis kernel $\mathbb{K}_{\mathcal{M}}(x, z) = \exp(-\psi((x - z)^\top \mathcal{M} (x - z)))$, one has

$$\nabla_z \mathbb{K}_{\mathcal{M}}(x, z) = -2\,\psi'\big((x - z)^\top \mathcal{M} (x - z)\big)\, \mathcal{M} (x - z)\, \mathbb{K}_{\mathcal{M}}(x, z). \tag{10}$$

Thus, the magnitude of gradients is directly determined by $\psi'(\cdot)$.

**Lemma 2** *For Laplace and Gaussian kernels with Mahalanobis distance, the gradient magnitudes satisfy*

$$\begin{aligned}
\|\nabla_z \mathbb{K}_{\mathcal{M}}^{\mathrm{Lap}}(x, z)\| &\propto \|\mathcal{M}(x - z)\|\, \mathbb{K}_{\mathcal{M}}^{\mathrm{Lap}}(x, z), \\
\|\nabla_z \mathbb{K}_{\mathcal{M}}^{\mathrm{Gauss}}(x, z)\| &\propto \|\mathcal{M}(x - z)\|\,(x - z)^\top \mathcal{M} (x - z)\, \mathbb{K}_{\mathcal{M}}^{\mathrm{Gauss}}(x, z).
\end{aligned} \tag{11}$$

The Laplace kernel yields gradient magnitudes that scale linearly in the Mahalanobis distance, whereas the Gaussian kernel introduces a quadratic scaling factor. Empirically, the linear scaling produces sharper and more stable local sensitivities, which improves the quality of AGOP estimates. Moreover, the Laplace form integrates the metric $M$ transparently, simplifying the recursive update and reducing the risk of overly flat or degenerate gradients. For these reasons, the Mahalanobis Laplace kernel offers a favorable balance between expressivity and numerical stability in RFM updates.

## B   DETAILS OF TASKS

We list the detailed tasks in `MMLU-Medical`, `MMLU-Financial`, `MMLU-Math`, and `MMLU-Humanities` as follows:

- `MMLU-Medical`: It contains six tasks: `Anatomy`, `Clinical Knowledge`, `College Biology`, `College Medicine`, `Medical Genetics`, `Professional Medicine`.
- `MMLU-Financial`: It contains three tasks: `Econometrics`, `High School Macroeconomics`, `High School Microeconomics`.
- `MMLU-Math`: It contains four tasks: `Abstract Algebra`, `College Mathematics`, `Elementary Mathematics`, `High School Mathematics`.
- `MMLU-Humanities`: It contains twelve tasks: `Formal Logic`, `Global Facts`, `High School European History`, `High School US History`, `High School World History`, `Human Aging`, `Logical Fallacies`, `Moral Disputes`, `Moral Scenarios`, `Philosophy`, `Prehistory`, `World Religions`.

To assess the safety of the LLMs, we follow (Dubey et al., 2024) and evaluate the performance with a fine-tuned harmful classifier based on the DeBERTaV3.[1]Moreover, we use SacreBLEU to evaluate the performance on the `Flores-en2zh` and `Flores-zh2en` tasks.

## C  BASELINES

To validate the effectiveness of our method, we select the followig methods as baselines:

- **Supervised Fine-Tuning (SFT):** We fine-tune all parameters of LLMs using the AdamW optimizer with a learning rate of $1 \times 10^{-5}$ and a batch size of 8. This process is conducted over three epochs on 2 NVIDIA A100 GPUs (80GB). During training, we use a linear learning rate schedule with a warm-up phase that constitutes 10% of the total training steps.
- **Knowledge Distillation (KD):** We use the expert model as the teacher and the LLMs as the student. The student model is trained on the instruction-tuning training set of each domain with the knowledge distillation loss. Boizard et al. (2024) proposes a method designed to facilitate knowledge distillation between teacher models and student models by leveraging optimal transport theory to enable distillation across models with different architectures and tokenizers.
- **Inference-Time Intervention (ITI):** Li et al. (2023) operates by modifying the activations of specific attention heads during inference. ITI identifies a subset of attention heads within the model that exhibit high linear probing accuracy for the classification of positive answers and the corresponding negative answers. During inference, activations are shifted along directions calculated based on the linear probes.
- **Contrastive Activation Addition (CAA):** Rimsky et al. (2024) computes steering vectors by averaging the difference in the hidden states between pairs of positive and negative examples. During inference, these steering vectors are added at all token positions after the user's prompt with either a positive or negative coefficient, allowing precise control over the degree of the targeted behavior.
- **Semantic-Adaptive Dynamic Intervention (SADI):** Wang et al. (2024c) dynamically generates steering vectors tailored to each input's semantic context. SADI first computes activation differences between positive and negative pairs, which are then used to create a binary mask that highlights the most impactful model components. During inference, SADI applies the binary mask to the user input activations, scaling them element-wise based on the input's semantic direction, thereby dynamically steering the model's behavior.
- **Sparse Activation Steering (SAS):** Bayat et al. (2025) proposes to derive steering vectors directly from the sparse representations learned by sparse autoencoders, allowing these vectors to be selectively amplified or suppressed as needed. By leveraging contrastive prompt-pairing, SAS is able to identify interpretable, behaviour-specific features, thereby enabling precise control over both the enhancement and suppression of targeted behaviors. For ITI, CAA, SADI, and SAS, we extract steering vectors using the development set of each task to build the necessary contrastive pairs.

## D  TRAINING DETAILS

We explore two knowledge-transfer scenarios. In the first, we transfer from a domain-specific expert model (e.g., medical, financial, mathematical) to a general-purpose target model by training auto-encoders on 2,000 domain-specific examples, using 500 domain-specific examples for mutual information analysis to identify intervention layers, and then employing 2,000 domain-specific examples as positive inputs alongside 2,000 general-domain examples as negative inputs to train RFMs. In the second scenario, we transfer from a larger general-purpose expert model to a smaller general-purpose target model. Similarly, we train the auto-encoders on 2,000 general-domain examples, using 500 general-domain examples to identify intervention layers. We then training RFMs on 2,000 general-domain examples as positive inputs and 2,000 domain-specific (e.g., medical) examples as negative inputs. All experiments are conducted on a single A100 GPU with 40 GB of memory.

---

[1]https://huggingface.co/domenicrosati/deberta-v3-xsmall-beavertails-harmful-qa-classifier

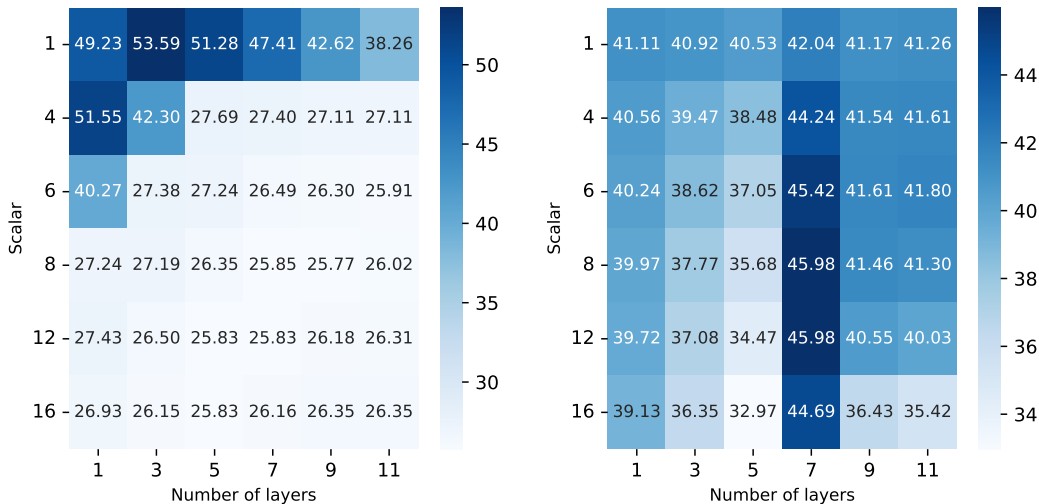

Figure 5: The selection of the number of intervention layers and scalar with Llama-3.1-8B-Instruct on the MedQA task.

Figure 6: The selection of the number of intervention layers and scalar with Qwen2.5-7B-Instruct on the MedQA task

## E  MORE ANALYSES

In this section, we include more analyses on on hyperparameters (Section E.1), model sizes (Section E.2), kernel types (Section E.3), and few-shot prompting (Section E.4).

### E.1  HYPERPARAMETER SELECTION

In Figure 5 and Figure 6, we sweep two hyperparameters to control the intervention: the number of intervention layers and the scalar. The number of intervention layers indicates how many layers we intervene in the model, and the scalar is used to control the strength of the intervention. Results indicate that the optimal settings for these hyperparameters vary across different models. This variability underscores that for precise task performance optimization, it is recommended to search for optimal hyperparameters using data from the validation sets with a small volume.

### E.2  RESULTS OF DIFFERENT MODEL SIZES

EXPERTSTEER delivers larger performance gains with smaller models. We further investigate the effectiveness of EXPERTSTEER across varying model sizes. We conduct experiments with the Llama series (Llama-3.1-8B-Instruct, Llama-3.2-3B-Instruct, Llama-3.2-1B-Instruct) and present the results in Figure 7. EXPERTSTEER consistently improves performance across all the model sizes. Notably, we observe that EXPERTSTEER yields larger performance gains in smaller models. This trend can be attributed to the fact that smaller models have a limited capacity to store knowledge, making them benefit more from external interventions like EXPERTSTEER.

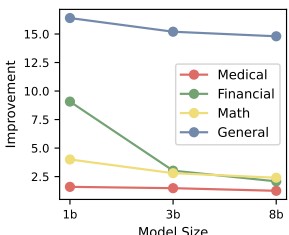

Figure 7: Performance gains of EXPERTSTEER across various model sizes on four domains.

### E.3 RESULTS OF OTHER KERNELS

As discussed in Section 3.3, we implement RFMs with the Laplacian kernel. In this section, we further investigate the effectiveness of EXPERTSTEER with other kernels, including the Gaussian kernel and the Linear kernel. As shown in Table 7, we find that RFMs with the Laplacian kernel consistently outperforms other kernels across all tasks. This indicates that the Laplacian kernel is more effective in extracting the knowledge from the expert model, validating the effectiveness of our design choice.

Table 7: Comparisons of different kernels used in RFMs on the Llama-3.1-8B-Instruct.

| Kernel | $\mu_{ALL}$ | MedQA | Med MCQA | MMLU Med. |
|---|---|---|---|---|
| baseline | 52.00 | 45.60 | 49.40 | 60.99 |
| Laplacian | 56.98 | 53.59 | 50.66 | 66.71 |
| Gaussian | 53.86 | 46.98 | 50.83 | 63.77 |
| Linear | 53.41 | 47.31 | 49.55 | 63.36 |

### E.4 RESULTS OF FEW-SHOT PROMPTING

We also conduct experiments to evaluate the performance of EXPERTSTEER with few-shot setting. The results of 5-shots prompting are shown in Table 8. While EXPERTSTEER improves the performances in few-shot setting, the gains are less pronounced compared to those in zero-shot setting. This suggests that few-shot examples already provide a strong learning signal (to both Baseline and EXPERTSTEER), somewhat overshadowing the additional benefits derived from steering vectors.

Table 8: Improvement with 5-shot prompting on the Qwen2.5-7B-Instruct backbone.

| | MMLU Med. | MMLU Fin. | MMLU Math. | MMLU Hum. |
|---|---|---|---|---|
| 0-shot | | | | |
| Baseline | 61.50 | 62.49 | 26.75 | 63.41 |
| EXPERTSTEER | 67.53 | 71.00 | 31.17 | 66.44 |
| 5-shot | | | | |
| Baseline | 74.86 | 76.79 | 54.43 | 76.45 |
| EXPERTSTEER | 76.09 | 77.56 | 54.92 | 77.14 |

## F TIME CONSUMED ANALYSIS FOR EACH STEP

As illustrated in Figure 4, the total time required to obtain steering vectors is approximately 17 minutes. We provide a detailed breakdown of the time consumption for each step in Table 9. When the data volume is small, the identification of layer pairs (step 2) constitutes the primary bottleneck, as it requires computing mutual information across all possible layer pairs. However, as the data volume increases, the time required for training RFMs (step 3) grows linearly and eventually exceeds that of layer pair identification. This is because training RFMs involves calculating the gradient outer product for every data point in each iteration, making its time cost directly proportional to the data volume.

Table 9: Time consumed (in seconds) for each step with the increasing number of training examples.

| #examples | total time | step 1 time | step 2 time | step 3 time |
|---|---|---|---|---|
| 200 | 313 | 6 | 258 | 49 |
| 400 | 470 | 7 | 378 | 85 |
| 600 | 533 | 8 | 380 | 145 |
| 800 | 571 | 9 | 387 | 175 |
| 1000 | 653 | 12 | 388 | 253 |
| 1200 | 687 | 12 | 390 | 285 |
| 1400 | 750 | 13 | 408 | 329 |
| 1600 | 878 | 13 | 422 | 443 |
| 1800 | 936 | 16 | 436 | 484 |
| 2000 | 1024 | 16 | 458 | 550 |

Table 12: Comparison between different feature extraction methods on the medical tasks and general tasks.

| | Llama-3.1-8B-Instruct | | | Qwen2.5-7B-Instruct | | | Gemma-2-2b-Instruct | | |
|---|---|---|---|---|---|---|---|---|---|
| | Med MCQA | NLI | ARC-C | Med MCQA | NLI | ARC-C | Med MCQA | NLI | ARC-C |
| Baseline | 49.40 | 57.87 | 67.39 | 46.25 | 71.00 | 73.54 | 33.06 | 41.82 | 38.17 |
| EXPERTSTEER | | | | | | | | | |
| ├ MD | 48.89 | 59.03 | 67.49 | 47.43 | 72.47 | 73.69 | 33.11 | 42.05 | 38.31 |
| ├ PCA | 48.87 | 59.08 | 67.57 | 48.19 | 73.94 | 75.34 | 33.13 | 42.33 | 38.41 |
| └ RFM | **50.66** | **61.36** | **68.34** | **48.57** | **77.20** | **78.24** | **33.37** | **44.10** | **39.11** |

## G  LAYER PAIRS SELECTION

As discussed in Section 3.2, our approach selects layer pairs with the lowest mutual information for intervention, aiming to maximize the effectiveness of knowledge transfer. In this section, we compare this strategy with two alternatives: selecting layer pairs based on the largest mean difference (MD) and randomly selecting layer pairs (Random). All three methods select the same number of layer pairs. As shown in Table 10, the mutual information-based selection consistently outperforms the other methods across all tasks. Notably, random selection yields only marginal improvements, highlighting that not all layers are equally suitable as knowledge sources or intervention points. These results demonstrate that mutual information is a more reliable and principled criterion for identifying the most impactful layer pairs, thereby validating the effectiveness of our design choice.

Table 10: Comparisons of different methods of slecting layer pairs on the Llama-3.1-8B-Instruct.

| Kernel | $\mu_{ALL}$ | MedQA | Med MCQA | MMLU Med. |
|---|---|---|---|---|
| Baseline | 52.00 | 45.60 | 49.40 | 60.99 |
| Random | 52.70 | 45.96 | 49.80 | 62.34 |
| MD | 53.51 | 46.31 | 50.05 | 64.18 |
| MI | 56.98 | 53.59 | 50.66 | 66.71 |

## H  SUPPLEMENTARY RESULTS OF FEATURE EXTRACTION METHODS

We have demonstrated the effectiveness of EXPERTSTEER with RFM in Section 5.1 using the Llama-3.1-8B-Instruct and Qwen2.5-7B-Instruct backbones. In this section, we further validate the effectiveness of EXPERTSTEER with other feature extraction methods on the Gemma-2-2b-Instruct backbone. As shown in Table 11, EXPERTSTEER with RFMs outperforms PCA and MD across all tasks. This is consistent with the results in Section 5.1, which indicate that RFMs are more effective than simple linear feature extraction methods. Furthermore, we also provide comparisons on the MedMCQA, NLI, and ARC-C tasks across three models in Table 12. The results show that EXPERTSTEER with RFMs consistently outperforms other feature extraction methods across all tasks.

Table 11: Comparison between different feature extraction methods on the medical tasks and general tasks.

| | MedQA | MMLU Med. | COPA | MMLU Hum. |
|---|---|---|---|---|
| Gemma-2-2b-Instruct | | | | |
| Baseline | 28.63 | 31.81 | 72.32 | 34.36 |
| EXPERTSTEER | | | | |
| ├ MD | 28.69 | 32.06 | 72.83 | 34.38 |
| ├ PCA | 28.77 | 32.34 | 72.91 | 34.39 |
| └ RFMs | **29.39** | **33.87** | **75.57** | **34.63** |

## I  SUPPLEMENTARY RESULTS OF VECTOR GENERATION SOURCE

The steering vectors used in previous studies are extracted from the model itself Li et al. (2023); Rimsky et al. (2024), but we argue that the steering vectors should be more effective if they are generated by expert models. In this section, we investigate the effectiveness of using steering vectors generated from the model itself (Self-generated) and those generated from expert models (Expert-generated). As shown in Table 13, we find that the steering vectors generated from expert models are more effective than those generated from the model itself. This indicates that the steering vectors generated from expert models can better capture additional knowledge and improve the performance

Table 13: Comparisons between steering vectors generated from model itself and expert model.

| | Medical | | | | NLU | | | | |
| --- | --- | --- | --- | --- | --- | --- | --- | --- | --- |
| | $\mu_{ALL}$ | MedQA | Med MCQA | MMLU Med. | $\mu_{ALL}$ | COPA | NLI | ARC-C | MMLU Hum. |
| **Llama-3.1-8B-Instruct** | | | | | | | | | |
| Baseline | 52.00 | 45.60 | 49.40 | 60.99 | 64.68 | 74.01 | 57.87 | 67.39 | 59.45 |
| Self-generated | 53.60 | 48.30 | 49.51 | 62.98 | 65.29 | 76.48 | 57.81 | 67.39 | 59.49 |
| Expert-generated | 56.98 | 53.59 | 50.66 | 66.71 | 68.45 | 83.47 | 61.36 | 68.34 | 60.60 |
| **Qwen2.5-7B-Instruct** | | | | | | | | | |
| Baseline | 49.65 | 41.20 | 46.25 | 61.50 | 72.51 | 82.07 | 71.00 | 73.54 | 63.41 |
| Self-generated | 50.08 | 41.30 | 46.74 | 62.22 | 78.52 | 94.17 | 80.12 | 75.22 | 64.60 |
| Expert-generated | 54.03 | 45.98 | 48.57 | 67.53 | 77.53 | 88.23 | 77.20 | 78.24 | 66.44 |
| **Gemma-2-2b-Instruct** | | | | | | | | | |
| Baseline | 31.17 | 28.63 | 33.06 | 31.81 | 46.67 | 72.32 | 41.82 | 38.17 | 34.36 |
| Self-generated | 31.17 | 28.64 | 33.09 | 31.79 | 47.03 | 73.41 | 42.15 | 38.22 | 34.35 |
| Expert-generated | 32.21 | 29.39 | 33.37 | 33.87 | 48.35 | 75.57 | 44.10 | 39.11 | 34.63 |

of EXPERTSTEER. These findings are consistent with the results in Figure 2 in Section 5.1, which show that expert models provide more effective guidance for generation.

## J  RESULTS OF PRESERVING PERFORMANCE EVALUATION

We further investigate the generalization ability of EXPERTSTEER by evaluating its performance on general-domain tasks following its intervention in the medical domain. As shown in Table 14, EXPERTSTEER retains competitive accuracy on these tasks, indicating that its adaptation to the specialized medical context does not compromise its foundational competencies. These results suggest that EXPERTSTEER can effectively transfer the expertise gained from the medical domain while preserving knowledge relevant to more general tasks. Moreover, EXPERTSTEER not only maintains its perfor-

Table 14: Performance on the general domain tasks after intervention in the medical domain with Llama-3.1-8B-Instruct. The expert model is Bio-Medical-Llama-3-8B.

| | $\mu_{ALL}$ | COPA | NLI | ARC-C | MMLU Hum. |
| --- | --- | --- | --- | --- | --- |
| Expert Model | 63.63 | 70.10 | 55.29 | 71.71 | 57.42 |
| Baseline | 64.68 | 74.01 | 57.87 | 67.39 | 59.45 |
| EXPERTSTEER | 64.73 | 74.10 | 56.26 | 72.40 | 63.16 |

mance but also outperforms the baseline in certain scenarios. This trend underscores the capacity of EXPERTSTEER to leverage complementary insights from both the specialized expert model and the original model. These findings highlight the robustness of our intervention strategy and its potential to scale across diverse application settings.

## K  SUPPLEMENTARY RESULTS OF GENERAL DOMAIN TASKS

We supply additional results of SFT and KD on the tasks of general domain in Table 15.

Table 15: Fine-tuning baselines in the general domain on the NLU tasks and Safety tasks with Llama-3.1-8B-Instruct, Qwen2.5-7B-Instruct, and Gemma-2-2b-Instruct models.

| | NLU | | | | | Safety | | |
|---|---|---|---|---|---|---|---|---|
| | $\mu_{\text{ALL}}$ | COPA | NLI | ARC-C | MMLU Hum. | $\mu_{\text{ALL}}$ | Salad | Harm Behav. |
| **Llama-3.1-8B-Instruct** | | | | | | | | |
| SFT | 69.29 | 92.62 | 51.16 | 73.48 | 59.90 | 67.40 | 65.60 | 69.20 |
| KD | 69.11 | 91.33 | 50.87 | 72.37 | 61.86 | 64.60 | 64.20 | 65.00 |
| **Qwen2.5-7B-Instruct** | | | | | | | | |
| SFT | 82.01 | 95.92 | 76.16 | 83.72 | 72.26 | 75.80 | 76.00 | 75.60 |
| KD | 83.57 | 96.58 | 77.22 | 85.87 | 74.62 | 79.90 | 77.60 | 82.20 |
| **Gemma-2-2b-Instruct** | | | | | | | | |
| SFT | 65.07 | 88.86 | 46.48 | 67.22 | 57.72 | 79.70 | 78.20 | 81.20 |
| KD | 57.33 | 74.62 | 42.12 | 58.95 | 53.61 | 78.70 | 76.20 | 81.20 |

