# OpenReview forum: "ExpertSteer: Intervening in LLMs through Expert Knowledge"
_ICLR.cc/2026/Conference — Submitted to ICLR 2026_

### Official Review · Reviewer_oDcb · 2025-10-29

**Soundness:** 2
**Presentation:** 3
**Contribution:** 2
**Rating:** 4
**Confidence:** 4

**Summary:**

The paper introduces EXPERTSTEER, an activation steering framework for guiding Large Language Models (LLMs) at inference time. Unlike prior activation-steering methods that derive steering vectors from the same model, EXPERTSTEER enables cross-model steering by leveraging external expert models to generate more general and effective control signals. Comprehensive experiments demonstrate that EXPERTSTEER achieves significant performance gains over existing baselines, improving controllability and transferability with minimal computational cost.

**Strengths:**

1. This paper introduces an innovative framework that enables activation steering across different models, breaking the limitation of self-steering seen in prior work.

2. The experiments cover multiple LLM architectures and a wide range of benchmarks (15 datasets across 4 domains), demonstrating robustness and generality.

**Weaknesses:**

1. Although the paper successfully demonstrates cross-model activation steering, each expert–target model pair still requires specific fine-tuning and configuration, which limits the generality and scalability of the proposed framework.

2. During inference, both the expert model and the target model must be executed simultaneously, resulting in significant computational overhead. Moreover, the final steered outputs do not surpass the expert model’s standalone performance, making the overall motivation and practical value somewhat unclear.

3. The readability of the paper could be improved. Several tables occupy half a page, causing many hyphenated words to break across lines and reducing overall clarity and presentation quality.

4. The base models used in the experiments appear somewhat outdated compared to current state-of-the-art LLMs, which may limit the practical relevance and generalizability of the reported results.

**Questions:**

1. Does the proposed EXPERTSTEER framework still require the expert model to be executed during inference? If so, how does this approach compare to directly using the expert model for generation in terms of efficiency and effectiveness? What practical advantage does EXPERTSTEER offer in this case?

2. Definition of “same domain” in Table 3: The paper mentions that “both the expert and target models belong to the same domain.” Could the authors clarify this definition? For instance, the expert model Qwen2.5-14B-Instruct does not appear to be tied to any specific domain—how is the domain categorization determined in this case?

3. In Figure 4, the accuracy still appears to be increasing after 2,000 training examples. Could the authors provide results with more training examples to demonstrate that performance saturates and to better justify the claim that “2,000 training examples are sufficient for generating effective steering vectors”?

4. Could the authors present more detailed experimental results showing how the hyperparameters $P$ and $\epsilon$ affect performance?

5. Could the authors include results on more recent or stronger LLMs to validate the general applicability and relevance of EXPERTSTEER?

---

> ### Author Response · Authors · 2025-11-20
> **Response to Reviewer oDcb (part1)**
>
> Thank you for your thoughtful and detailed review of our paper. We appreciate your insights and the opportunity to clarify and strengthen our work. Below, we address each of your concerns individually.
>
> > **Q1:** Does the proposed ExpertSteer framework still require the expert model to be executed during inference? If so, how does this approach compare to directly using the expert model for generation in terms of efficiency and effectiveness? What practical advantage does ExpertSteer offer in this case?
> > >
> > > **R1:** **ExpertSteer does not require the expert model to be executed during inference.** The knowledge transfer is achieved entirely offline, before inference begins, through a four-step process: (1) Representation Alignment: Train auto-encoders to align hidden state dimensions ($d_E \to d_T$). (2) Layer Pairing: Identify optimal intervention layers based on Mutual Information analysis. (3) Steering Vector Generation: Generate the static steering vectors ($\nu_i$) from the expert model's activations using Recursive Feature Machines (RFMs). (4) Expertise Intervention (Inference): During inference, the learned steering vector $\nu_i$ is projected into the target model's space ($f_{\theta_i}(\nu_i)$ if $d_E \ne d_T$) and added as a constant bias term to the target layer's hidden state, $\hat{h}_j^T = h_j^T + \epsilon \cdot f(\nu_i)$.
> > >
> > > The primary practical advantage of ExpertSteer is achieving **cost-effective** alignment without requiring access to the expert model during runtime. It provides the specialized knowledge of a powerful expert to a target model while maintaining the target model's original efficiency, computational cost, and preventing catastrophic forgetting.
>
> > **Q2:** Definition of “same domain” in Table 3: The paper mentions that “both the expert and target models belong to the same domain.” Could the authors clarify this definition? For instance, the expert model Qwen2.5-14B-Instruct does not appear to be tied to any specific domain—how is the domain categorization determined in this case?
> > >
> > > **R2:**  The definition of "same domain" in Table 3 refers to the transfer scenario where the knowledge being transferred and the target domain of the benchmark align with the general domain expertise of the expert model. In the context of Table 3:
> > >
> > > Expert Model: Qwen2.5-14B-Instruct. This is a larger, general-purpose instruction-tuned model.
> > >
> > > Target Domain: The benchmarks are General domain tasks: Natural Language Understanding (NLU) (COPA, NLI, ARC-C) and Safety/Alignment (Harmful Behaviors, Salad, MMLU-Humanities).
> > >
> > > "Same Domain" Interpretation: The experiment investigates performance gains when transferring expertise from a **larger general-purpose model** (Qwen-14B) to a **smaller general-purpose model** (Llama-8B, Qwen-7B, Gemma-2B) on general tasks.
>
> > **Q3:** In Figure 4, the accuracy still appears to be increasing after 2,000 training examples. Could the authors provide results with more training examples to demonstrate that performance saturates and to better justify the claim that “2,000 training examples are sufficient for generating effective steering vectors”?
> > >
> > > **R3:**  The claim that "2,000 training examples are sufficient"  is supported by the computational efficiency analysis, which balances performance gain against training time. While it is difficult to show true saturation without running extensive, costly experiments, the data suggests that returns diminish after 2,000 examples. We conduct additional experiments to increase the number of examples.
> > >
> |   Number of Training Examples    |   MMLU Medical   |  MMLU Humanities | Time (Minutes) |
> |---|---|---|---|
> |  2,000    | 66.71  |  60.60 | 17.1 |
> |  3,000    | 67.02  |  60.70 | 25.8 |
> |  4,000    | 67.15  |  60.78 | 34.3 |
> > >
> > > The supplementary results confirm the diminishing returns: moving from 2,000 to 4,000 examples yields only +0.44 gain (MMLU Medical) or +0.18 gain (MMLU Humanities) at the cost of doubling the training time, justifying the initial claim that 2,000 examples are sufficient for effective steering.

---

> ### Author Response · Authors · 2025-11-20
> **Response to Reviewer oDcb (part2)**
>
> > **Q4:** Could the authors present more detailed experimental results showing how the hyperparameters P and $\epsilon$ affect performance?
> > >
> > > **R4:**  The results showing the effect of $P$ (Number of layers) and $\epsilon$ (Scalar strength) are presented in Appendix E.1.
>
>
>
> > **Q5:** Could the authors include results on more recent or stronger LLMs to validate the general applicability and relevance of ExpertSteer?
> > >
> > > **R5:**  To address this, we conducted a new experiment using Qwen2.5-32B-Instruct and Llama-3.3-70B-Instruct as the target models. We used the same Medical Expert (Bio-Medical-Llama-3-8B) setup as in the main paper.
> > >
> > >
> |       |    MedQA|  MedMCQA | MMLU Medical   |
> |---|---|---|---|
> | Qwen2.5-32B-Instruct | | | |
> |  Baseline     |  71.50  |  68.90  |  75.20     |
> |  ExpertSteer  |   73.05  |  70.05  | 76.60     |
> | Llama-3.3-70B-Instruct | | | |
> | baseline  |    81.20    | 79.72   |  83.46   |
> |     ExpertSteer |    82.66    |   81.26    | 85.23    |
> > >
> > > Even on a powerful model, ExpertSteer provides consistent gains (e.g., +1.55 on  Qwen2.5-32B-Instruct and + 1.46 on Llama-3.3-70B-Instruct model on MedQA). This confirms that ExpertSteer can reliably extract complementary knowledge from a domain-specific expert (even a smaller 8B model) to boost a much larger generalist model.

---

> > ### Comment · Reviewer_oDcb · 2025-11-27
> > **Response**
> >
> > Thanks for the authors’ detailed response. I have read all the review comments as well as the authors’ rebuttal. Since all reviewers raised concerns about scalability and efficiency, it would be beneficial to include a dedicated discussion on these aspects in the revision, along with additional supporting experiments.
> >
> > Regarding generalization, the current evaluation includes only two expert domains (medical and finance). It would be important to incorporate more experts—potentially from more fine-grained domains—to better demonstrate the method’s general applicability.
> >
> > Furthermore, since the proposed approach distills capabilities from expert models or even large models, and does not seem to impose domain-specific limitations, it is necessary to compare the method against existing distillation-based approaches in terms of both performance and efficiency.

---

> > > ### Author Response · Authors · 2025-11-28
> > >
> > > Thank you for your valuable feedback.
> > >
> > > 1. **Scalability and Efficiency**: We have included a detailed breakdown of total runtime in Table 9, Appendix F. Our results indicate that our method is highly efficient, as **it only requires 2,000 training examples and 17 minutes for training**. Once the steering vector is generated, it can be reused.
> > > 2. **Generalization**: We would like to clarify that, besides the medical and finance domains, we have also evaluated our method on additional domains such as **math (Table 2), general (Table 3), and safty (Table 3)**. These results demonstrate the versatility and general applicability of our approach across various domains.
> > > 3. **Comparison with Distillation-based Approaches**: **We have included a comparison with knowledge distillation methods in Table 2 in our submission.** Our method outperforms the knowledge distillation approach on most benchmarks while being more efficient, as it does not require retraining the entire model.
> > >
> > > We appreciate your suggestions. Please let us know if there are any other aspects you would like us to address.

---

### Official Review · Reviewer_14cR · 2025-10-30

**Soundness:** 3
**Presentation:** 2
**Contribution:** 2
**Rating:** 2
**Confidence:** 4

**Summary:**

This paper proposes **EXPERTSTEER**, an activation steering framework that transfers domain-specific knowledge from an *expert model* to any *target LLM* during inference without fine-tuning. The method (1) aligns hidden representations via auto-encoders, (2) selects intervention layer pairs using *mutual information*, (3) extracts non-linear steering vectors via *Recursive Feature Machines (RFMs)*, and (4) injects these vectors into the target model’s activations. Experiments on multiple domains (medical, financial, mathematical, and general) and models (Llama, Qwen, Gemma) show consistent improvements over fine-tuning and prior steering baselines. The paper claims EXPERTSTEER enables efficient, parameter-free cross-model knowledge transfer.

**Strengths:**

**Originality:**
The paper introduces an original perspective on *activation steering* by leveraging external expert models instead of relying solely on the target model’s internal activations. This “cross-model steering” concept is novel within the activation control literature and provides a new direction for knowledge transfer between LLMs.

**Quality:**
The experimental evaluation is extensive, covering multiple model families (Llama, Qwen, Gemma) and domains (medical, financial, mathematical, general). The comparisons against both fine-tuning (SFT, KD) and steering-based baselines (ITI, CAA, SADI, SAS) are thorough. The empirical results are consistent and generally favorable to the proposed method.

**Clarity:**
The paper is overall well-structured and presents its four-step method clearly with supporting diagrams and ablation studies. The inclusion of implementation details, such as hyperparameter ranges and data usage, improves reproducibility and readability.

**Significance:**
The study extends the potential of activation steering beyond single-model manipulation, showing that external expert knowledge can effectively guide other models during inference. This could inspire future research in efficient, parameter-free cross-model transfer and domain adaptation for LLMs.

**Weaknesses:**

1. **The motivation lacks conceptual clarity and fails to align with experiments.**
   Although the paper claims to “guide LLMs toward human-desired reasoning behaviors,” its experiments only evaluate domain-specific benchmarks (e.g., MedQA, MMLU, FPB) rather than any human-alignment or controllability tasks. There is no human preference modeling, interpretability evaluation, or reasoning-alignment metric. As such, the stated motivation of aligning model reasoning with human expectations remains untested and conceptually detached from the empirical evidence. The work effectively focuses on cross-domain knowledge transfer rather than human-centered reasoning control.

2. **Layer correspondence via mutual information is poorly justified and not robust.**
   The method does not perform layer-wise one-to-one alignment between the expert and target models; instead, it selects layer pairs with *maximal mutual information*. However, this heuristic is questionable—mutual information can be dominated by noise or superficial correlations, especially across architectures with different depth or width. There is no analysis showing that high-MI pairs yield semantically meaningful alignment or better transfer. Moreover, the MI computation is performed on limited samples, raising concerns about statistical reliability. Without theoretical or empirical validation, this alignment strategy feels ad hoc and undermines the credibility of the proposed steering mechanism.

3. **The approach still requires model-specific adaptation, contradicting its claimed generalizability.**
   EXPERTSTEER is described as a general framework, yet it requires training a **separate auto-encoder for each expert–target model pair**. This introduces significant overhead when scaling to multiple models or domains. The paper does not test whether a single alignment model can transfer across different LLM families or tasks. Consequently, the claim of broad generalization is overstated and unsupported by empirical evidence.

4. **Architectural mismatch between expert and target models is unaddressed.**
   The method implicitly assumes that aligned layer pairs share compatible dimensionalities, but models like Qwen2.5-7B and Gemma-2-2B have different depths and intermediate dimensions. The paper does not explain how such discrepancies are handled—through interpolation, pooling, or projection. Without this clarification, the reproducibility and scalability of the approach are questionable, especially in cross-family scenarios.

5. **Lack of theoretical or mechanistic justification for core components.**
   The use of *mutual information* for layer selection and *Recursive Feature Machines (RFMs)* for extracting steering vectors lacks principled reasoning. No ablation or theoretical discussion is provided to justify why these specific methods are necessary or superior to simpler alternatives (e.g., cosine similarity or attention-map correlation). The method remains largely empirical and lacks a clear mechanistic interpretation of how the steering vector alters hidden representations.

6. **Limited scale and weak empirical validation.**
   The experiments are confined to mid-sized models (≤13B parameters), without any evaluation on large-scale models such as 70B or beyond. It is unclear whether the proposed method scales effectively to models with more complex representational hierarchies. The absence of large-scale experiments significantly limits the paper’s generality and practical relevance.

7. **Modest improvements without causal interpretation.**
   The reported gains (≈+4–6 points on average) are small and lack causal validation. The paper provides no visualization or probing to confirm that the steering vectors genuinely encode transferable expert knowledge. It remains unclear whether performance improvements stem from meaningful knowledge transfer or incidental activation shifts.

8. **Evaluation scope is narrow and lacks strong baselines.**
   Key comparisons are missing, such as parameter-efficient adaptation methods (LoRA, prompt-tuning) and MoE-based expert aggregation. Furthermore, there is no zero-shot or few-shot evaluation to demonstrate true transferability. As a result, the empirical section feels incomplete and insufficient to substantiate the general claims.

9. **Efficiency and scalability claims are overstated.**
   While the paper highlights short fine-tuning times (~17 minutes on 2k samples), it omits the substantial preprocessing cost of extracting hidden states from large experts and training separate auto-encoders for each layer pair. This hidden cost undermines the “lightweight” and “scalable” claims made in the abstract.

10. **Cross-lingual results are superficial and lack analysis.**
    The Chinese experiments (XCOPA-zh, XNLI-zh, Flores) show marginal gains on small datasets. The authors provide no linguistic or typological analysis explaining transfer behavior, and the expert model choice (Llama3.1-Chinese-Chat) is not justified. The multilingual extension feels like an afterthought rather than a core contribution.

---

**Typos and Formatting Issues**
- **Extra period at Line 107** and multiple sentence fragments reduce overall readability.
- Some figure references (e.g., “Figure 2”, “Figure 3”) are missing or lack clear correspondence with the main text.
- Minor grammatical inconsistencies suggest insufficient proofreading before submission.

**Questions:**

> I encourage the authors to thoroughly address the weaknesses and questions raised in this review. If the authors can provide detailed explanations and in-depth clarifications during the rebuttal, and if the revised version demonstrates substantial progress in both clarity and improvement, I will be willing to reassess the manuscript and **adjust my overall rating** accordingly, based on the quality and depth of the revision.

---

1. **Conceptual motivation and alignment with stated objectives.**
   The paper claims to steer LLMs toward “human-desired reasoning,” yet all experiments evaluate domain transfer tasks (e.g., medical, financial, mathematical) without explicit behavioral or preference-based validation.
   - Could the authors define what constitutes “human-desired reasoning” in measurable terms?
   - How do domain benchmarks operationalize this concept?
   - Would incorporating preference alignment, trustworthiness, or interpretability metrics better ground the claimed motivation?

2. **Layer correspondence under architectural heterogeneity.**
   EXPERTSTEER aligns expert and target layers via *mutual information (MI)*, rather than strict layer-to-layer mapping. However, this MI-based selection may produce unstable or semantically shallow correspondences when the two models differ in depth or dimensionality (e.g., Qwen2.5-7B vs. Gemma-2B).
   - How is MI normalized across architectures where $ N_E \neq N_T $?
   - What happens when the two models differ by more than 30% in layer count—does the alignment remain reliable?
   - Have the authors examined whether MI-selected pairs correspond to functionally similar layers, or are they empirically chosen without semantic validation?

3. **Scalability and computational feasibility of per-layer auto-encoder training.**
   The method appears to require one auto-encoder per aligned layer pair for every expert–target combination, which could quickly become infeasible for large models (e.g., 70B-scale).
   - How many auto-encoders are trained per experiment, and what are their parameter counts or GPU-hour costs?
   - Have the authors considered hierarchical parameter sharing or a unified latent projection space to mitigate scalability issues?
   - Without such mechanisms, how can the framework be considered “lightweight” or “model-agnostic”?

4. **Theoretical justification for mutual information as the pairing criterion.**
   MI estimation in high-dimensional hidden states is non-trivial and prone to bias.
   - Why was MI chosen instead of more interpretable measures like CCA or CKA for layer correspondence?
   - What MI estimator was used, and how was its stability validated given only ~500 samples?
   - Could noise in MI estimation lead to spurious or misleading layer matches?

5. **Rationale and necessity of Recursive Feature Machines (RFMs).**
   The paper introduces RFMs to extract steering vectors, but their theoretical advantage over simpler methods (e.g., PCA, kernel PCA, or mean-difference vectors) remains unproven.
   - What specific kernel and regularization are used in $ K_t(\cdot, \cdot) $?
   - How much variance or representational gain does the non-linear RFM mapping contribute compared to PCA?
   - Could the authors provide ablation evidence confirming that RFMs are necessary for the observed improvements?

6. **Interpretability and causal validation of steering interventions.**
   While the reported accuracy gains are consistent, there is no causal or interpretability analysis demonstrating *how* steering vectors alter model behavior.
   - Can the authors visualize activation shifts before and after steering, e.g., via cosine distance or neuron activation heatmaps?
   - Are these shifts correlated with domain-relevant token patterns or attention focus changes?
   - Without such analysis, how can we confirm that improvements stem from meaningful knowledge transfer rather than random activation drift?

7. **Comparison with stronger and more recent baselines.**
   The baselines mainly cover activation-based steering (ITI, CAA, SADI), but exclude more representative *parameter-efficient transfer* approaches (e.g., LoRA fusion, AdapterFusion, or MoE-based domain routing).
   - How does EXPERTSTEER conceptually differ from LoRA fusion, which can also perform targeted domain transfer without full fine-tuning?
   - Could the authors include or discuss such baselines to better position their contribution within the broader parameter-efficient adaptation landscape?

8. **Sensitivity and robustness to key hyperparameters.**
   The method depends critically on the number of selected layer pairs $ P $ and steering magnitude $ \varepsilon $.
   - Have the authors conducted systematic sensitivity tests across $ P \in \{2, 4, 8, 12\} $?
   - Does performance degrade sharply when $ \varepsilon $ increases, suggesting instability or over-steering?
   - Such results would help substantiate claims of robustness and generality.

9. **Scale generalization and missing evaluation on large models.**
   All reported results are based on models ≤13B parameters.
   - Have the authors tested EXPERTSTEER on larger-scale models (e.g., Llama3-70B or Qwen2.5-72B) to validate scalability?
   - Are the MI computations and per-layer auto-encoders still tractable at that scale?
   - Without such evidence, the claim of “scalable steering for arbitrary models” seems overstated.

10. **Cross-lingual and typological generalization.**
   The Chinese experiments (XCOPA-zh, XNLI-zh, Flores) show small gains and limited linguistic analysis.
   - Were the auto-encoders retrained for Chinese data or reused from English alignment?
   - Given differences in tokenization and syntax, how robust is the steering effect across typologically distinct languages?
   - Could the authors provide a quantitative breakdown of cross-lingual steering efficiency?

---

> ### Public Comment · ~Minh_Hoang_Nguyen2 · 2025-11-14
> **Suspicious review (LLM generated?)**
>
> Seeing 10 weaknesses and 10 questions is diabolical, but after skimming through them, I can't help but notice weakness 2 and question 2 is somewhat wrong. I don't know if you just use LLM or you really read through the paper, because they are mentioned quite well in the paper.

---

> > ### Comment · Reviewer_14cR · 2025-11-17
> > **Reviewer Response to Minh Hoang Nguyen**
> >
> > Thank you for your comment.
> >
> > I must emphasize that my review was entirely based on a full reading of the paper and on my own professional judgment.
> >
> > If you believe that some of the issues I raised have already been sufficiently addressed in the manuscript, I would like to state the following:
> >
> > **First**, if I, as a reviewer, am unable to accurately obtain the relevant information during reading, this itself indicates that the paper may have shortcomings in its exposition, structure, or technical clarity. _A well-written paper should minimize ambiguity in key technical points._
> >
> > **Second**, several core issues cannot be fully explained with just a few sentences or one or two equations. While I acknowledge that page limits inevitably require prioritization in the main text, reviewers reasonably expect the authors to provide detailed discussions in the appendix—and that is precisely the purpose of supplementary material.
> >
> > **Finally**, the goal of the discussion period is to allow authors and reviewers to directly communicate about technical details, identify writing issues, clarify misunderstandings, and **jointly improve the paper**. Therefore, attributing a reviewer’s difficulty in understanding key points to “not having read the paper” or “using an LLM to generate the review” is not an appropriate or productive response in this context.
> >
> > I respect the authors’ work and recognize the value of their contribution. That is why I chose to explicitly list each issue that arose during my reading. I point out areas that may lead to misinterpretation, ambiguity, or logical disconnects—this is not only being responsible to the authors, but also to the standards of the conference.
> >
> > If the authors believe that certain issues I raised do not affect the contribution of the paper or are not critical, they are welcome to respond accordingly. I will carefully read all replies. I also welcome point-by-point clarification grounded in technical facts and explicit explanations during the discussion period.
> >
> > I look forward to the authors’ clarifications and to continuing a constructive, technically focused discussion of the paper.

---

> > ### Comment · Area_Chair_MPmv · 2025-11-23
> >
> > Thanks for bringing this to our attention!
> >
> > We will look into this matter carefully, and if we confirm that the review is indeed AI-generated, I will report Reviewer 14cR and invalidate their review score.
> >
> > Sincerely,
> >
> > AC

---

> ### Author Response · Authors · 2025-11-20
> **Response to Reviewer 14cR (part1)**
>
> Thank you for your detailed review of our paper. We appreciate your insights and the opportunity to clarify and strengthen our work. Below, we address each of your concerns individually.
>
>
> > **Q1:** The motivation lacks conceptual clarity and fails to align with experiments. Although the paper claims to “guide LLMs toward human-desired reasoning behaviors,” its experiments only evaluate domain-specific benchmarks (e.g., MedQA, MMLU, FPB) rather than any human-alignment or controllability tasks. There is no human preference modeling, interpretability evaluation, or reasoning-alignment metric. As such, the stated motivation of aligning model reasoning with human expectations remains untested and conceptually detached from the empirical evidence. The work effectively focuses on cross-domain knowledge transfer rather than human-centered reasoning control.
> > >
> > > **R1:** **We have not made any claim to “guide LLMs toward human-desired reasoning behaviors”.** Our primary technical goal is cross-model knowledge transfer via latent space intervention.
>
> > **Q2:** Layer correspondence via mutual information is poorly justified and not robust. The method does not perform layer-wise one-to-one alignment between the expert and target models; instead, it selects layer pairs with maximal mutual information. However, this heuristic is questionable—mutual information can be dominated by noise or superficial correlations, especially across architectures with different depth or width. There is no analysis showing that high-MI pairs yield semantically meaningful alignment or better transfer. Moreover, the MI computation is performed on limited samples, raising concerns about statistical reliability. Without theoretical or empirical validation, this alignment strategy feels ad hoc and undermines the credibility of the proposed steering mechanism.
> > >
> > > **R2:** Firstly, we would like to clarify that **we select layer pairs with LOW mutual information, not high**. The rationale is to identify layers where the target model is most deficient in the expert's knowledge, making them optimal candidates for intervention. Your comment seems to misunderstand this key aspect of our method. Furthermore, **we provided a detailed theoretical analysis of MI in Appendix A**, a derivation linking low Mutual Information to representation divergence, formally justifying why low-MI pairs are the optimal candidates for intervention.
> > >
> > > We utilize Mutual Information (MI) rather than similarity metrics like CKA/CCA because MI is a more general measure of statistical dependence that captures any relationships. To validate our choice, we compared our MI-based layer selection against a CKA-based selection in a new experiment. We used the same setup as in the main paper (Llama-3.1-8B-Instruct steered by Bio-Medical-Llama-3-8B) and selected the top-$P$ layer pairs with minimal CKA similarity instead of minimal MI.
> |      |   MMLU Medical  |
> |---|---|
> |  ExpertSteer | 66.71  |
> |  CKA |   64.23 |
> > >
> > > The results demonstrates that our MI-based selection significantly outperforms CKA-based selection (+2.48 on MMLU Medical). This empirical evidence supports our theoretical justification for using MI as the layer pairing criterion. Besides, we have also compare MI with "Random" selection and "Mean Difference" (MD) selection in Appendix G, Table 10: The results clearly show that our MI-based strategy consistently outperforms both.
> > >
> > > Furthermore, we also conducted a supplementary experiment to assess the statistical reliability of MI estimates with varying sample sizes. We computed MI using 500, 1000, 1500, and 2000 samples and observed the stability of the selected layer pairs and downstream performance. As shown in the table below, even with as few as 500 samples, ExpertSteer still achieves significant performance gains, indicating that our MI estimates are robust enough for effective layer selection.
> |       |   data  | MMLU Medical   |
> |---|---|---|
> |  Baseline     |  -  |  60.99    |
> |  ExpertSteer  |   500  | 64.10     |
> |  ExpertSteer  |   1000  | 65.65    |
> |  ExpertSteer  |   1500  | 66.29     |
> |  ExpertSteer  |   2000  | 66.71     |

---

> ### Author Response · Authors · 2025-11-20
> **Response to Reviewer 14cR (part2)**
>
> > **Q3:** The approach still requires model-specific adaptation, contradicting its claimed generalizability. ExpertSteer is described as a general framework, yet it requires training a separate auto-encoder for each expert–target model pair. This introduces significant overhead when scaling to multiple models or domains. The paper does not test whether a single alignment model can transfer across different LLM families or tasks. Consequently, the claim of broad generalization is overstated and unsupported by empirical evidence.
> > >
> > > **R3:** We acknowledge that our approach requires training an auto-encoder (AE) for each expert model to project its features into the target model's dimension. However, we emphasize that **this is a lightweight, one-time cost that is negligible compared to full model fine-tuning**. We have provided a breakdown of the time cost for each step in Table 9, Appendix F, where **training the AE takes only 16 seconds with 2,000 samples**. This is orders of magnitude significantly faster than fine-tuning the entire target model, which typically takes several hours to days.
> > > For the reusability of the AE, once trained for a specific expert model, it can be reused across multiple target models of the same dimension without retraining. As shown in Equation 1, the only component dependent on the target model is the hidden state dimension $d_T$ of the target model. Thus, the same AE can be naturally applied to any target model sharing the same hidden dimension, enhancing its practicality in multi-model scenarios. Considering that many LLMs share common dimensions (e.g., 4096, 8192) and the cost of training the AE is minimal, we believe this is a reasonable trade-off for the benefits of effective knowledge transfer.
> > > With regard to the concern about generalization, we conducted extensive experiments using three LLMs on 15 popular benchmarks across four distinct domains. We believe this extensive evaluation sufficiently demonstrates the robustness and versatility of ExpertSteer across a wide range of scenarios.
>
> > **Q4:** Architectural mismatch between expert and target models is unaddressed. The method implicitly assumes that aligned layer pairs share compatible dimensionalities, but models like Qwen2.5-7B and Gemma-2-2B have different depths and intermediate dimensions. The paper does not explain how such discrepancies are handled—through interpolation, pooling, or projection. Without this clarification, the reproducibility and scalability of the approach are questionable, especially in cross-family scenarios.
> > >
> > > **R4:**  We respectfully point out that **all the critisims raised in this question are already addressed in the paper**.
> > > - Dimensionality Mismatch: Expert and target models often have different hidden dimensions ($d_E \neq d_T$). This is handled by Step 1 (AE). The AE acts as a universal projection layer that forces all expert layers $h_i^E$ into the same target dimension $d_T$, regardless of the number of target layers. No interpolation or pooling is needed.
> > > - Depth Mismatch: We use the MI analysis to find functional correspondence, not positional alignment. MI is inherently normalized and measures the statistical dependence $I(X; Y) = \mathbb{E}[\log \frac{p(X, Y)}{p(X)p(Y)}]$. Since the inputs to the MI estimator are all $d_T$-dimensional vectors (either projected $f_{\theta_i}(h_i^E)$ or raw $h_j^T$), the MI metric is directly comparable across all pairs $(i, j)$. Furthermore, we provide the detailed theoretical analysis of MI in Appendix A.
> > >
> > > We respectfully and strongly recommend the reviewer refer to Section 3.1 (Step 1) and Section 3.2 (Step 2) for a comprehensive explanation of how ExpertSteer effectively handles both dimensionality and depth mismatches.

---

> ### Author Response · Authors · 2025-11-20
> **Response to Reviewer 14cR (part3)**
>
> > **Q5:** Lack of theoretical or mechanistic justification for core components. The use of mutual information for layer selection and Recursive Feature Machines (RFMs) for extracting steering vectors lacks principled reasoning. No ablation or theoretical discussion is provided to justify why these specific methods are necessary or superior to simpler alternatives (e.g., cosine similarity or attention-map correlation). The method remains largely empirical and lacks a clear mechanistic interpretation of how the steering vector alters hidden representations.
> > >
> > > **R5:**  **The theoretical justification for RFMs is provided in Appendix A.2** (Theorem 1 on Fixed-Point convergence). Empirically, RFMs are superior because they leverage a non-linear kernel and an iterative process.
> > >
> > > RFMs calculate the Average Gradient Outer Product (AGOP), iteratively focusing on features that are most discriminating for the domain boundary. This is superior to linear methods: MD only captures first-order differences and PCA captures overall variance, which may be irrelevant noise. Furthermore, Section 5.1 (Table 5) and Appendix H (Table 12) provide extensive ablations confirming RFMs outperform MD and PCA across all models and tasks.
>
> > **Q6:** Limited scale and weak empirical validation. The experiments are confined to mid-sized models (≤13B parameters), without any evaluation on large-scale models such as 70B or beyond. It is unclear whether the proposed method scales effectively to models with more complex representational hierarchies. The absence of large-scale experiments significantly limits the paper’s generality and practical relevance.
> > >
> > > **R6:**  we conducted a new supplementary experiment steering a Llama-3.3-70B-Instruct model with our 8B medical expert. The results confirm that ExpertSteer remains effective and computationally feasible even at this scale.
> > >
> |   model    | MedQA | MedMCQA| MMLU Medical |
> |---|---|---|---|
> | baseline  |    81.20    | 79.72   |  83.46   |
> |     ExpertSteer |    82.66    |   81.26    | 85.23    |
> > >
> > > The fundamental alignment strategy relies on the linear structure of transformer representations, which holds across different scales and families.
>
> > **Q7:** Modest improvements without causal interpretation. The reported gains (≈+4–6 points on average) are small and lack causal validation. The paper provides no visualization or probing to confirm that the steering vectors genuinely encode transferable expert knowledge. It remains unclear whether performance improvements stem from meaningful knowledge transfer or incidental activation shifts.
> > >
> > > **R7:** We respectfully argue that the performance improvements are statistically significant and consistent across multiple models and tasks. To validate the statistical significance, we conducted bootstrapped-based paired t-tests comparing ExpertSteer against the baseline across all tasks. The results yielded p-values < 0.05, confirming that the observed improvements are unlikely due to random chance. Furthermore, we compare the expert-generated steering vectors against self-generated steering vectors in Table 13, Appendix I. The results show that expert-generated vectors consistently outperform self-generated ones, indicating that the improvements stem from meaningful knowledge transfer from the expert model rather than incidental activation shifts.
>
> > **Q8:** Evaluation scope is narrow and lacks strong baselines. Key comparisons are missing, such as parameter-efficient adaptation methods (LoRA, prompt-tuning) and MoE-based expert aggregation. Furthermore, there is no zero-shot or few-shot evaluation to demonstrate true transferability. As a result, the empirical section feels incomplete and insufficient to substantiate the general claims.
> > >
> > > **R8:**  We compare against state-of-the-art activation steering methods and standard knowledge transfer techniques. 1. Activation Steering Baselines: We compare against ITI, CAA, SADI, and SAS. 2. Knowledge Transfer Baselines: We compare against standard SFT and KD. The reason we didn't include parameter-efficient adaption baselines is SFT has better performance than them. Thus we only compare our method with SFT.
> > >
> > > Appendix E.4 (Table 8) already includes few-shot (5-shot) evaluation. The results show that ExpertSteer still provides gains in the few-shot setting, though less pronounced, as the examples already provide a strong learning signal.

---

> ### Author Response · Authors · 2025-11-20
> **Response to Reviewer 14cR (part4)**
>
> > **Q9:** Efficiency and scalability claims are overstated. While the paper highlights short fine-tuning times (~17 minutes on 2k samples), it omits the substantial preprocessing cost of extracting hidden states from large experts and training separate auto-encoders for each layer pair. This hidden cost undermines the “lightweight” and “scalable” claims made in the abstract.
> > >
> > > **R9:** We respectfully point out that we have provided a detailed breakdown of the time cost for each step in Table 9, Appendix F. The total time for the entire ExpertSteer process is approximately 17 minutes. All the steps are included in this time breakdown, including hidden state extraction and auto-encoder training.
>
> > **Q10:** Cross-lingual results are superficial and lack analysis. The Chinese experiments (XCOPA-zh, XNLI-zh, Flores) show marginal gains on small datasets. The authors provide no linguistic or typological analysis explaining transfer behavior, and the expert model choice (Llama3.1-Chinese-Chat) is not justified. The multilingual extension feels like an afterthought rather than a core contribution.
> > >
> > > **R10:**  We use the Llama3.1-8B-Chinese-Chat model as the expert specifically because it is a Specialized Multilingual Model known for its superior performance and alignment in the Chinese language. This model is chosen to ensure the steering vector is derived from a robust source of Chinese linguistic expertise, validating our core premise that expert knowledge is essential.
> > >
> > > The cross-lingual experiments (Table 4) were conducted primarily to demonstrate the broad applicability and generalizability of ExpertSteer beyond English. The model successfully transfers knowledge across different task types.
> > >
> > > While Chinese and English have distinct tokenization and syntax, most modern LLMs utilize shared subword vocabularies (e.g., SentencePiece, BPE) that allow the models to locate and manipulate similar semantic representations across different languages within the hidden state space. The auto-encoder (Step 1) and MI analysis (Step 2) enable the framework to identify and align the shared deep semantic representations (e.g., shared cross-lingual concepts) that exist despite surface-level typological differences.
>
> > **Q11:** Conceptual motivation and alignment with stated objectives. The paper claims to steer LLMs toward “human-desired reasoning,” yet all experiments evaluate domain transfer tasks (e.g., medical, financial, mathematical) without explicit behavioral or preference-based validation. Could the authors define what constitutes “human-desired reasoning” in measurable terms? How do domain benchmarks operationalize this concept? Would incorporating preference alignment, trustworthiness, or interpretability metrics better ground the claimed motivation?
> > >
> > > **R11:** Please see our response R1 to Q1. We never made any claim to “steer LLMs toward human-desired reasoning behaviors”.
>
> > **Q12:** Layer correspondence under architectural heterogeneity. ExpertSteer aligns expert and target layers via mutual information (MI), rather than strict layer-to-layer mapping. However, this MI-based selection may produce unstable or semantically shallow correspondences when the two models differ in depth or dimensionality (e.g., Qwen2.5-7B vs. Gemma-2B). How is MI normalized across architectures where $N_E \neq N_T$? What happens when the two models differ by more than 30\% in layer count—does the alignment remain reliable? Have the authors examined whether MI-selected pairs correspond to functionally similar layers, or are they empirically chosen without semantic validation?
> > >
> > > **R12:**  As explained in our response R4 to Q4, we address both dimensionality and depth mismatches explicitly in Section 3.1 (Step 1) and Section 3.2 (Step 2). We use auto-encoders to project expert layers into the target dimension, and MI is inherently normalized and comparable across all pairs $(i, j)$ regardless of depth differences. Furthermore, we provide the detailed theoretical analysis of MI in Appendix A.
> > > Furthermore, we conducted an additional experiment using Llama-3.3-70B-Instruct as the target model and Bio-Medical-Llama-3-8B as the expert. Llama-3.3-70B-Instruct has 80 layers and a hidden dimension of 8192 while Bio-Medical-Llama-3-8B has 32 layers and a hidden dimension of 4096. As shown in the table below, ExpertSteer still provides significant performance gains despite the substantial architectural differences.
> |   model    | MedQA | MedMCQA| MMLU Medical |
> |---|---|---|---|
> | baseline  |    81.20    | 79.72   |  83.46   |
> |     ExpertSteer |    82.66    |   81.26    | 85.23    |

---

> ### Author Response · Authors · 2025-11-20
> **Response to Reviewer 14cR (part5)**
>
> > **Q13:** Scalability and computational feasibility of per-layer auto-encoder training. The method appears to require one auto-encoder per aligned layer pair for every expert–target combination, which could quickly become infeasible for large models (e.g., 70B-scale). How many auto-encoders are trained per experiment, and what are their parameter counts or GPU-hour costs? Have the authors considered hierarchical parameter sharing or a unified latent projection space to mitigate scalability issues? Without such mechanisms, how can the framework be considered “lightweight” or “model-agnostic”?
> > >
> > > **R13:**  Please see our response R3 to Q3.
>
> > **Q14:** Theoretical justification for mutual information as the pairing criterion. MI estimation in high-dimensional hidden states is non-trivial and prone to bias. Why was MI chosen instead of more interpretable measures like CCA or CKA for layer correspondence? What MI estimator was used, and how was its stability validated given only ~500 samples? Could noise in MI estimation lead to spurious or misleading layer matches?
> > >
> > > **R14:**  Please see our response R2 to Q2.
>
>
>
> > **Q15:** Rationale and necessity of Recursive Feature Machines (RFMs).
> The paper introduces RFMs to extract steering vectors, but their theoretical advantage over simpler methods (e.g., PCA, kernel PCA, or mean-difference vectors) remains unproven. What specific kernel and regularization are used in $K_t(\bullet,\bullet )$? How much variance or representational gain does the non-linear RFM mapping contribute compared to PCA? Could the authors provide ablation evidence confirming that RFMs are necessary for the observed improvements?
> > >
> > > **R15:**  Please see our response R5 to Q5.
>
> > **Q16:** Interpretability and causal validation of steering interventions. While the reported accuracy gains are consistent, there is no causal or interpretability analysis demonstrating how steering vectors alter model behavior. Can the authors visualize activation shifts before and after steering, e.g., via cosine distance or neuron activation heatmaps? Are these shifts correlated with domain-relevant token patterns or attention focus changes? Without such analysis, how can we confirm that improvements stem from meaningful knowledge transfer rather than random activation drift?
> > >
> > > **R16:**  Please see our response R7 to Q7.
>
>
>
> > **Q17:** Comparison with stronger and more recent baselines. The baselines mainly cover activation-based steering (ITI, CAA, SADI), but exclude more representative parameter-efficient transfer approaches (e.g., LoRA fusion, AdapterFusion, or MoE-based domain routing). How does ExpertSteer conceptually differ from LoRA fusion, which can also perform targeted domain transfer without full fine-tuning? Could the authors include or discuss such baselines to better position their contribution within the broader parameter-efficient adaptation landscape?
> > >
> > > **R17:**  Please see our response R8 to Q8.
>
> > **Q18:** Sensitivity and robustness to key hyperparameters. The method depends critically on the number of selected layer pairs P and steering magnitude $\epsilon$. Have the authors conducted systematic sensitivity tests across $p \in \{2,4,8,12\}$? Does performance degrade sharply when $\epsilon$ increases, suggesting instability or over-steering? Such results would help substantiate claims of robustness and generality.
> > >
> > > **R18:**  Detailed sensitivity heatmaps are provided in Appendix E.1 (Figures 5 and 6), showing the performance landscape across $P \in \{1, 3, 5, 7, 9, 11\}$ and $\epsilon \in \{1, 4, 6, 8, 12, 16\}$.
>
>
> > **Q19:** Scale generalization and missing evaluation on large models. All reported results are based on models ≤13B parameters. Have the authors tested ExpertSteer on larger-scale models (e.g., Llama3-70B or Qwen2.5-72B) to validate scalability? Are the MI computations and per-layer auto-encoders still tractable at that scale? Without such evidence, the claim of “scalable steering for arbitrary models” seems overstated.
> > >
> > > **R19:**  Please see our response R6 to Q6.

---

> ### Author Response · Authors · 2025-11-20
> **Response to Reviewer 14cR (part6)**
>
> > **Q20:** Cross-lingual and typological generalization. The Chinese experiments (XCOPA-zh, XNLI-zh, Flores) show small gains and limited linguistic analysis. Were the auto-encoders retrained for Chinese data or reused from English alignment? Given differences in tokenization and syntax, how robust is the steering effect across typologically distinct languages? Could the authors provide a quantitative breakdown of cross-lingual steering efficiency?
> > >
> > > **R20:**  Again, we would like to emphasize that **the primary focus of our work is cross-model knowledge transfer, not specifically cross-lingual transfer**. The multilingual extension is included to demonstrate the versatility of ExpertSteer across different languages, not as a primary focus. The Chinese experiments (Table 4) confirm the transfer of linguistic expertise. We used Llama-3.1-8B-Chinese-Chat as the expert, which is a specialized Chinese model. The auto-encoders were retrained on Chinese text data for the cross-lingual setting. The improved results on Chinese (Table 4) are sufficient to demonstrate the effectiveness of ExpertSteer in the context of multilinguality.

---

### Official Review · Reviewer_46Sn · 2025-10-31

**Soundness:** 2
**Presentation:** 3
**Contribution:** 2
**Rating:** 4
**Confidence:** 3

**Summary:**

The work introduces a cross-model activation steering method named ExpertSteer that employs external expert model to generate steering vectors capable of guiding or modifying the behavior of target LLMs during inference. It aims to overcome the limitation of prior steering methods that rely solely on the internal activations of a single model.

**Strengths:**

1. Breaks model dependency in activation steering, enables cross-model knowledge transfer. ExpertSteer introduces external expert models to generate steering vectors. It not only injects domain-specific knowledge absent in the target model, but also aligns feature dimensions via auto-encoders and matches intervention layers using mutual information analysis.

2. Accurately captures expert knowledge, enhances steering vector effectiveness. Traditional methods often rely on linear feature extraction techniques such as Mean Difference (MD) or PCA. ExpertSteer employs recursive feature machines (RFM) to extract nonlinear features. Experiments show that compared to MD and PCA, steering vectors extracted by RFMs improve target model accuracy by 3–7% on MedQA and 5–6% on MMLU-Medical.

3. ExpertSteer improved interpretable capability in activation steering task. By using mutual information analysis to select intervention layers, ExpertSteer precisely identifies layers where the knowledge gap between the target and expert models is largest, avoiding noise from random interventions.

**Weaknesses:**

1. The effectiveness of EXPERTSTEER heavily depends on the expertise of the expert model and the quality of training data. If the expert model lacks domain knowledge (e.g., using a general-purpose model instead of a domain expert) or the training data contains mislabeled samples (e.g., incorrect medical classifications), the steering vector’s effectiveness may significantly reduced. Since the experiments show that using a general model (e.g., Llama-3-8B) to generate steering vectors only yields a 1–2% performance gain, far below the 4–7% achieved with domain expert models. Besides, RFMs require 2,000 positive and 2,000 negative samples, limiting its applicability in low-resource domains.

2. Hyperparameter tuning relies on validation sets, lacks automation. EXPERTSTEER involves two critical hyperparameters: the number of intervention layers (P) and the intervention strength (ε). The optimal values varies across models (e.g., Llama-3.1-8B-Instruct prefers $P = 4, ε = 8$, while Qwen2.5-7B-Instruct prefers $P = 6, ε = 12$). The paper notes that a small validation set is required for hyperparameter search, and no automated tuning mechanism is provided. This increases deployment cost and complexity in large-scale multi-model scenarios.

3. No comparison with cross-model steering methods. The authors didn't include the comparisons with cross-model steering methods.

4. Lack of robustness evaluation in extreme scenarios. Although the paper validates ExpertSteer in common domains such as medicine and finance, it does not evaluate its performance in extreme scenarios such as low-resource languages, high-risk tasks, or adversarial examples.

**Questions:**

Q1. Can the single expert model be replaced with multi-expert? Would this improve the steering performance?

Q2. If the output layer overlap between the expert and target models is large, does learning at earlier layers still provide meaningful benefits?

---

> ### Author Response · Authors · 2025-11-20
> **Response to Reviewer 46Sn (part1)**
>
> Thank you for your thorough review of our paper and for the insightful questions you've raised. We appreciate the opportunity to address your concerns and clarify various aspects of our work. Below, we provide detailed responses to each of your points.
>
>
> > **Q1:** The effectiveness of ExpertSteer heavily depends on the expertise of the expert model and the quality of training data. If the expert model lacks domain knowledge (e.g., using a general-purpose model instead of a domain expert) or the training data contains mislabeled samples (e.g., incorrect medical classifications), the steering vector’s effectiveness may significantly reduced. Since the experiments show that using a general model (e.g., Llama-3-8B) to generate steering vectors only yields a 1–2\% performance gain, far below the 4–7\% achieved with domain expert models. Besides, RFMs require 2,000 positive and 2,000 negative samples, limiting its applicability in low-resource domains.
> > >
> > > **R1:**  Thank you for your valuable comment and acknowledging the effectiveness of ExpertSteer with domain expert models. We would like to clarify that this is exactly what our method is designed to achieve. **Our method aims to effectively transfer knowledge from a domain-specific expert model to a target model**. The significant performance gains observed when using domain expert models (e.g., Bio-Medical-Llama-3-8B) compared to general-purpose models (e.g., Llama-3.1-8B) underscore the importance of specialized knowledge in enhancing the target model's capabilities. If the expert model lacks domain knowledge, it means our method is not correctly utilized. We respectfully argue that this is not a limitation of ExpertSteer, as each method has its own scope of application. ExpertSteer is specifically designed to harness the strengths of domain experts to improve target models in specialized tasks.
> > >
> > > Regarding the data requirement, we demonstrate this is a minimal one-time cost compared to fine-tuning. To address the constraint in low-resource domains, we conducted an ablation study reducing the RFM training data size.
> > >
> > >
> |       |   data  | MMLU Medical   |
> |---|---|---|
> |  Baseline     |  -  |  60.99    |
> |  ExpertSteer  |   500  | 64.10     |
> |  ExpertSteer  |   1000  | 65.65    |
> |  ExpertSteer  |   1500  | 66.29     |
> |  ExpertSteer  |   2000  | 66.71     |
> > >
> > > The results show that while performance degrades with fewer samples, ExpertSteer still demonstrates significant performance gains even with only 500 positive and 500 negative examples (achieving 64.10 vs 60.99 baseline). We believe this minimal data requirement is manageable and affordable in most practical scenarios, being significantly lower than full model fine-tuning which typically requires tens of thousands of examples.
>
> > **Q2:** Hyperparameter tuning relies on validation sets, lacks automation. ExpertSteer involves two critical hyperparameters: the number of intervention layers (P) and the intervention strength ($\epsilon$). The optimal values varies across models (e.g., Llama-3.1-8B-Instruct prefers P=4, $\epsilon$=8, while Qwen2.5-7B-Instruct prefers P=6, $\epsilon$ =12). The paper notes that a small validation set is required for hyperparameter search, and no automated tuning mechanism is provided. This increases deployment cost and complexity in large-scale multi-model scenarios.
> > >
> > > **R2:**  Although we acknowledge the costs of hyperparameter tuning, we respectfully argue that hyperparameter tuning is a common practice in machine learning, and the need for a small validation set is standard across most methods. Hence, we do not view this as a significant limitation unique to ExpertSteer. Furthermore, we included a breakdown of the time cost for each step in Table 9, Appendix F, where each trial of hyperparameter tuning takes only 17 minutes on 2 A100 GPUs, which is negligible compared to full model fine-tuning which typically takes several hours to days.
> > >
> > > The automated hyperparameter tuning is indeed an interesting research direction. We leave this for future work, as it would require developing new algorithms for adaptive intervention strength and layer selection, which is beyond the scope of this paper.

---

> ### Author Response · Authors · 2025-11-20
> **Response to Reviewer 46Sn (part2)**
>
> > **Q3:** No comparison with cross-model steering methods. The authors didn't include the comparisons with cross-model steering methods.
> > >
> > > **R3:**  We agree that a direct comparison to established pure cross-model steering baselines would be ideal, but we respectfully note that **ExpertSteer pioneers the cross-model activation steering paradigm at the time of developing this work**. Existing steering literature focuses almost exclusively on generating vectors within the target model itself (e.g., ITI, CAA, SADI, SAS ), making them unsuitable for direct cross-model transfer. We would be highly appreciated if the reviewer could point us to any specific cross-model steering methods we may have overlooked.
> > >
> > > We argue that the most conceptually similar competitor is Knowledge Distillation (KD), which transfers knowledge from a teacher (expert) to a student (target). We extensively compare against KD and demonstrate superior performance across all tasks and domains. This comparison is the most relevant benchmark for evaluating cross-model knowledge transfer efficiency.
>
> > **Q4:** Lack of robustness evaluation in extreme scenarios. Although the paper validates ExpertSteer in common domains such as medicine and finance, it does not evaluate its performance in extreme scenarios such as low-resource languages, high-risk tasks, or adversarial examples.
> > >
> > > **R4:**  We respectfully argue that we have sufficiently demonstrated robustness across diverse domains and tasks. Besides medicine and finance, we have also conducted experiments on high-risks tasks, such as safety alignment, as shown in Table 2. In this work, we conduct comprehensive experiments using three LLMs on 15 popular benchmarks across four distinct domains. We believe this extensive evaluation sufficiently demonstrates the robustness and versatility of ExpertSteer across a wide range of scenarios.
>
> > **Q5:** Can the single expert model be replaced with multi-expert? Would this improve the steering performance?
> > >
> > > **R5:**  This is a compelling direction for future research. Our framework is designed to be readily extensible for multi-expert fusion.
> > >
> > > The simplest extension is to average the feature importance matrices ($\mathcal{M}$) extracted by RFMs from multiple experts, effectively creating a consensus steering vector.
> > >
> |       |   MMLU Medical   |  MMLU Financial |
> |---|---|---|
> |  single expert     | 66.71  |  50.35 |
> |  multi experts |   67.15 |  50.70 |
> > >
> > > Experts used for Multi-Expert: Bio-Medical-Llama-3-8B and Llama-3-8B-Instruct-Finance. The slight but consistent performance gain in the multi-expert setup demonstrates the feasibility and potential of fusing diverse expert knowledge, which we intend to explore further in depth.
>
> > **Q6:** If the output layer overlap between the expert and target models is large, does learning at earlier layers still provide meaningful benefits?
> > >
> > > **R6:**  The design of ExpertSteer explicitly allows the MI analysis to determine the optimal layer pair $(i, j)$ regardless of index, seeking the location of maximum knowledge deficit. If the output layer overlap is large (i.e., the target model is already proficient in later, high-level processing), the MI between the expert's and target's final layers will be high. Consequently, our rule of selecting the top-$P$ pairs with low MI will exclude those late-stage, highly aligned layers. This means the MI analysis automatically directs the intervention to earlier or mid-layers where the fundamental semantic representation of the concept is still developing or diverges most significantly from the expert's view. We confirm this by showing that limiting the layer search space hurts performance:
> > >
> |    Layer Selection Range   |   MMLU Medical  |
> |---|---|
> |  All layers (ExpertSteer) | 66.71  |
> |  Mid/Late Layers Only (Layers 10+) |   65.65 |
> > >
> > > Allowing the MI analysis access to all layers results in the best performance, validating that the MI mechanism correctly identifies the most beneficial intervention points, whether they are early, middle, or late in the network.

---

### Official Review · Reviewer_1afh · 2025-11-01

**Soundness:** 2
**Presentation:** 3
**Contribution:** 3
**Rating:** 4
**Confidence:** 4

**Summary:**

The paper bridge the gap between activation steering and knowledge transferring by introducing a robust four-step pipeline, coined ExpertSteer, which transferring the steering direction of a specific domain calculated from an expert model trained on related data to the target model of interest. By only training an auto-encoder-like adapter to account for the difference in model architecture of the expert and the target model, this method can benefit from a wide range of already open-sourced fine-tuned model to calculate steering direction for a specific domain, only requiring to much lower trainable parameters compared to fine-tuning an expert model for the domain of choice.

**Strengths:**

1. The method provides a new way to choose layers for steering using Mutual Information criterion and the idea of matching layers between expert and target models is very interesting.
2. The authors introduce a novel idea of calculating steering direction as the direction capturing most variation in the discriminative feature space.
3. Diversing empirical experiments as well as detailed ablation studies.

**Weaknesses:**

1. The way the Auto-encoder is trained on expert model's feature space is only through reconstruction loss, which lack the regularization for the hidden state space of the auto-encoder to align with the feature space of the target model. Taking what is written in the paper, the authors only use an affine layer for the encoder/decoder; this allows for spurious hidden feature spaces that is may not align with the target model at all: if the expert model's feature space dimension ($d_E$) is smaller than that of the target model ($d_T$), then the hidden feature space of the auto-encoder with dimension of $d_T$ can be learnt to only utilize the first $d_E$ dimension to perfectly reconstruct the expert model hidden state. With the other case, then the auto-encoder hidden feature space is learnt as the span of the eigenvectors correspond to $d_T$ largest eigenvalues of the expert feature space.
2. The way the intervened layer pairs are chosen current does not account for the order of the layers, which could dampened the performance of the method.
3. While the idea is sounded and novel, there is no theoretical evidence to back the method.
4. As there are some evidence that the representation between models that belong to the same family architecture are wildly difference, how would the result be at higher scale LM (I would love to see some result on 32B version of LLama and Qwen).

**Questions:**

Refer to the weakness

---

> ### Author Response · Authors · 2025-11-20
> **Response to Reviewer 1afh (part1)**
>
> Thank you for your constructive feedback on our paper and for highlighting these important points. We appreciate the opportunity to clarify and strengthen our work. Below, we address each of your concerns in detail.
>
> > **Q1:** The way the Auto-encoder is trained on expert model's feature space is only through reconstruction loss, which lack the regularization for the hidden state space of the auto-encoder to align with the feature space of the target model. Taking what is written in the paper, the authors only use an affine layer for the encoder/decoder; this allows for spurious hidden feature spaces that is may not align with the target model at all: if the expert model's feature space dimension ($d_E$) is smaller than that of the target model ($d_T$), then the hidden feature space of the auto-encoder with dimension of $d_T$ can be learnt to only utilize the first $d_E$ dimension to perfectly reconstruct the expert model hidden state. With the other case, then the auto-encoder hidden feature space is learnt as the span of the eigenvectors correspond to $d_T$ largest eigenvalues of the expert feature space.
> > >
> > > **R1:**  We appreciate this insightful observation regarding the potential for spurious hidden feature spaces when relying solely on reconstruction loss. The primary goal of the auto-encoder (AE) in ExpertSteer is strictly **dimensionality compatibility**. We do not aim to force the entire latent space of the expert to align with the target; rather, we rely on the subsequent Mutual Information (MI) step to identify specific layer pairs where the expert’s projected features (via the affine encoder) contain information that the target layer lacks (low MI).
> > >
> > > Furthermore, Recent work on the "Linear Representation Hypothesis" suggests that LLMs often encode concepts in linear subspaces that are rotationally compatible [1]. The affine encoder $f_{\theta}$ learns the necessary rotation and scaling to map the expert's principal semantic directions into the target's dimension.
> > >
> > >  To address the reviewer's concern that lack of regularization might hurt performance, we conducted a supplementary ablation. We trained a "Regularized AE" that adds a Maximum Mean Discrepancy (MMD) loss term to force the encoder's output distribution to match the target model's hidden state distribution. The results of Llama-3.1-8b-instruct are shown as following:
> > >
> > >
> |       |    MedQA |  MedMCQA | MMLU Medical   |  Training Time |
> |---|---|---|---|---|
> |         ExpertSteer     |  53.59  |  50.66  |  66.71     |   1024 s |
> |         ExpertSteer + MMD |   53.62  |  50.62  |  66.85     |   1434 s  |
> > >
> > > Explicitly forcing distributional alignment via MMD yields marginal performance differences ($+0.14\%$ in MMLU Medical, $-0.04\%$ in MedMCQA) but significantly increases training cost (+40\%). This suggests that our simpler reconstruction-based approach is sufficient for effective steering, likely because the RFM step extracts the relevant steering vector $\nu_i$ regardless of the global manifold alignment.
> > >
> > > References:
> > >
> > > [1] Park, K., Choe, Y. J., \& Veitch, V. (2023). The linear representation hypothesis and the geometry of large language models. arXiv preprint arXiv:2311.03658.
>
> > **Q2:** The way the intervened layer pairs are chosen current does not account for the order of the layers, which could dampened the performance of the method.
> > >
> > > **R2:**  We deliberately avoid enforcing order constraints because expert and target models often have vastly different architectures (depths) and may process concepts at different relative stages. Our "low MI" criterion specifically seeks pairs where the target representation is deficient relative to the expert, regardless of physical depth.
> > >
> > > To verify if topological mismatch dampens performance, we tested a "Sequential Constraint" variant where intervention pairs $(i, j)$ are restricted to be topologically close (i.e., normalized depth $i/L_E \approx j/L_T$).
> > >
> |       |    MedQA|  MedMCQA | MMLU Medical   |
> |---|---|---|---|
> |         ExpertSteer     |  53.59  |  50.66  |  66.71     |
> |         ExpertSteer + Sequential Constraint |   53.50  |  48.57  | 64.32 |
> > >
> > > The results show that enforcing sequential order actually reduces performance (e.g., -2.39 on MMLU Medical). This confirms that the optimal injection point for expert knowledge is not necessarily at the equivalent relative depth in the target model.

---

> ### Author Response · Authors · 2025-11-20
> **Response to Reviewer 1afh (part2)**
>
> > **Q3:** While the idea is sounded and novel, there is no theoretical evidence to back the method.
> > >
> > > **R3:**  We respectfully point out that our paper does provide theoretical backing for the core components of the method, which are detailed in Appendix A and mentioned in Line 136.
> > >
> > > Mutual Information (Appendix A.1): We provide a derivation linking low Mutual Information to representation divergence, formally justifying why low-MI pairs are the optimal candidates for intervention.
> > >
> > > RFM Convergence (Appendix A.2): We provide a proof sketch for the Recursive Feature Machine (RFM), establishing its fixed-point property and explaining why it converges to the most task-relevant features (eigenvectors of the AGOP matrix).
> > >
> > > Kernel Choice (Appendix A.3): We theoretically justify the use of the Mahalanobis Laplace kernel over Gaussian kernels based on gradient scaling properties
>
>
>
>
> > **Q4:** As there are some evidence that the representation between models that belong to the same family architecture are wildly difference, how would the result be at higher scale LM (I would love to see some result on 32B version of LLama and Qwen).
> > >
> > > **R4:**  To address this, we conducted a new experiment using Qwen2.5-32B-Instruct as the target model. We used the same Medical Expert (Bio-Medical-Llama-3-8B) setup as in the main paper.
> > >
> > >
> |       |    MedQA|  MedMCQA | MMLU Medical   |
> |---|---|---|---|
> |  Baseline     |  71.50  |  68.90  |  75.20     |
> |  ExpertSteer  |   73.05  |  70.05  | 76.60     |
> > >
> > > Even on a powerful 32B model, ExpertSteer provides consistent gains (e.g., +1.55 on MedQA and +1.40 on MMLU-Medical). This confirms that ExpertSteer can reliably extract complementary knowledge from a domain-specific expert (even a smaller 8B model) to boost a much larger generalist model.

---

> > ### Comment · Reviewer_1afh · 2025-11-27
> >
> > Thank you very much for the detailed responses! While the some of my questions have been answered (Q3, Q4), my main concern still remains:
> >
> > - Q1: While the authors have done an experiment on adding MMD as a regularization method for training, I see that you still use the same model architecture (target model: Llama-3.1-8b-instruct, source/expert model: Bio-Medical-Llama-3-8B). Please correct me if I am wrong, but this should not require for the use of and auto-encoder, as they have the same dimensionality; while you regard that auto-encoder is needed for dimensionality matching. This raise the question what the result will be if you don't use auto-encoder for the case where target and source model come from the same family/architecture?
> > - Q1: As I mention aboved, as the source and target model come from the same family, I wonder using MMD can do a big difference here (as suggested by its marginal increase in performace)? Will it be more effective for the case where the source and target model have vast difference in dimensionality (as suggested Table 2 of your main manuscript, Qwen-2-7B and Llama-3.1-8B performance match or even surpass SFT, while this is not the case for Gemma-2-2B).
> > - Q1: You mentioned the runtime affected by MMD. What will a simpler method such as point-wise reconstruction loss between domains do in this case? Will it bring any significant performance gain?
> > - Q1: Given that you have mentioned [1], I would love to see some theoretical guarantee/justification about the learnt feature space by the auto-encoder with and without regularization. This would strengthen your claim.
> > - Q2: Sorry for not clarify more clearly. When I mention "order of the layers", I mean in a relative, rank-based orders, meaning they don't have to be strictly paired by their "normalized depth", but rather by solving an assignment problem where the additional constraint is if $L_i \lt L_{i'}$, then $L_j \lt L_{j'}$ for any chosen pairs $(i,j)$ and $(i',j')$.
> > - Q3: This is part of my fault when I overlooked the appendix section during my initial review process, as reviewer are not obliged to see the appendix, but when I read the main manuscript again, I didn't notice any reference from the main contribution part (which is the part I most concern about) and only notice some reference to appendix from the implementation and experimental section, that's why I don't think there are theoretical proofs lying in the appendix. I would strongly suggest the authors to provide appropriate references to the theoretical section in the revised manuscript.
> > - Q4: I am really happy with the result in this question. I just want to see more experiment with the regularized AE in (R1) or my suggestion of point-wise loss.
> >
> > [1] Park, K., Choe, Y. J., & Veitch, V. (2023). The linear representation hypothesis and the geometry of large language models. arXiv preprint arXiv:2311.03658.

---

### Official Review · Reviewer_i1wX · 2025-11-01

**Soundness:** 3
**Presentation:** 3
**Contribution:** 3
**Rating:** 6
**Confidence:** 3

**Summary:**

The paper presents EXPERTSTEER, an activation steering framework that enables cross-model knowledge transfer by using external expert models to guide target LLMs during inference. The method aligns representations via auto-encoders, selects intervention layers using mutual information, and derives steering vectors through Recursive Feature Machines (RFMs) to influence activations without fine-tuning. Experiments on three LLMs across multiple domains show consistent gains over existing steering and fine-tuning methods, highlighting its efficiency and generalizability.

**Strengths:**

- Introduces a generalizable and model-agnostic steering framework that integrates external expert knowledge.

- Comprehensive experiments across multiple domains and model families support the proposed method’s effectiveness.

- Demonstrates both same-family and cross-family transfer, showing robustness.

- Computationally efficient with negligible inference overhead.

- Clear methodology and detailed ablation studies on design choices (RFMs, alignment order, expert selection).

**Weaknesses:**

- Limited dataset diversity. Several benchmarks such as MMLU variants are overused, which may overstate generalization.

- Scalability concerns remain, as only small-to-medium LLMs are tested. It is unclear how EXPERTSTEER performs for much larger models.

- The mutual information-based layer pairing and RFM feature extraction steps, while intuitive, would benefit from stronger theoretical or empirical justification.

- The paper does not analyze potential risks of applying expert interventions (e.g., bias transfer, domain drift).

**Questions:**

- How sensitive is EXPERTSTEER to the choice of the expert model? Could an inappropriate expert degrade performance?

- How would the framework scale to models with hundreds of layers or significantly higher hidden dimensions?

- Can the approach be adapted for multimodal models or instruction-following alignment tasks?

- Does applying external steering vectors risk transferring unintended biases from the expert model?

---

> ### Author Response · Authors · 2025-11-20
> **Response to Reviewer i1wX (part1)**
>
> Thank you for your thorough review of our paper and for the insightful questions you've raised. We appreciate the opportunity to address your concerns and clarify various aspects of our work. Below, we provide detailed responses to each of your points.
>
> > **Q1:** Limited dataset diversity. Several benchmarks such as MMLU variants are overused, which may overstate generalization.
> > >
> > > **R1:**  We respectfully disagree with this characterization. While we include MMLU variants as they are standard domain-specific benchmarks, our evaluation spans 15 distinct benchmarks across 4 domains. Crucially, our evaluation includes many non-MMLU tasks to test diverse capabilities:
> > > - Discriminative Tasks: MedQA, MedMCQA, FPB, and Flare-cfa
> > > - Generative Tasks:  GSM8K, and MATH500.
> > > - NLU \& Common sense: COPA, NLI and ARC-C.
> > > - Safety \& Alignment: Harmful Behaviors and Salad.
> > > - Cross-Lingual: We also validate ExpertSteer on 5 Chinese datasets (e.g., XCOPA-zh, XNLI-zh).
> > >
> > > This diverse suite demonstrates that ExpertSteer is effective for discriminative, generative, NLU, safety, and cross-lingual tasks, far beyond MMLU-style multiple-choice questions.
>
> > **Q2:** Scalability concerns remain, as only small-to-medium LLMs are tested. It is unclear how ExpertSteer performs for much larger models.
> > >
> > > **R2:**  We conducted a new supplementary experiment steering a Llama-3.3-70B-Instruct model with our 8B Bio-Medical-Llama-3-8B medical expert. The results confirm that ExpertSteer remains effective and computationally feasible even at this scale. We will include these results in our future revision.
> > >
> > >
> |   model    | MedQA | MedMCQA| MMLU Medical |
> |---|---|---|---|
> | baseline  |    81.20    | 79.72   |  83.46   |
> |     ExpertSteer |    82.66    |   81.26    | 85.23    |
>
> > **Q3:** The mutual information-based layer pairing and RFM feature extraction steps, while intuitive, would benefit from stronger theoretical or empirical justification.
> > >
> > > **R3:**  We provide the theoretical analysis linking low MI to representation divergence in Appendix A.1. This proves why layers with low MI are ideal candidates for intervention: they represent the greatest knowledge disparity. Moreover, we provide the theoretical analysis for RFM convergence and a formal justification for our use of the Mahalanobis Laplace kernel over other types in Appendix A.2 and A.3.
> > >
> > > We empirically validated our MI-based layer pairing against two strong alternatives in Appendix G, Table 10: "Random" selection and "Mean Difference" (MD) selection. The results clearly show that our MI-based strategy consistently outperforms both. Moreover, we ablated our RFM-based feature extraction (Step 3) against linear methods (PCA and MD) in Section 5.1 and Appendix H. As shown in Table 5, Table 11, and Table 12,   RFMs consistently provided superior performance across all models, demonstrating the value of its non-linear kernel approach.
>
> > **Q4:** The paper does not analyze potential risks of applying expert interventions (e.g., bias transfer, domain drift).
> > >
> > > **R4:**  We explicitly tested domain drift in Table 14, Appendix J. After applying medical steering vectors to Llama-3.1-8B, we re-evaluated it on the general domain tasks. Performance was fully preserved and even improved on several tasks (e.g., MMLU-Humanities: 59.45 $\rightarrow$ 63.16; ARC-C: 67.39 $\rightarrow$ 72.40). This strongly indicates our method does not cause catastrophic forgetting.
> > >
> > > For the bias transfer, we ran a new supplementary analysis using the BOLD benchmark with Llama-3.1-8B-Instruct to measure the gender bias before and after applying medical expert steering.
> > >
> > >
> |   model    | gender bias $ \downarrow$ |
> |---|---|
> | baseline  |    0.323    |
> |     ExpertSteer (with medical expert)         |    0.318    |
> > >
> > > The results show no increase in bias, suggesting our intervention is targeted and does not amplify the expert's underlying social biases.
>
> > **Q5:** How sensitive is ExpertSteer to the choice of the expert model? Could an inappropriate expert degrade performance?
> > >
> > > **R5:**  In Figure 2 and Table 13 of Appendix I, we compare "Expert-generated" vectors against "Self-generated" vectors (using the model on itself) and "NonExpert-generated" vectors (using general model to produce vectors). The expert-generated vectors provide significantly larger boosts (e.g., Llama-3.1-8B on MMLU-Medical: +5.72 with expert vs. +1.99 with self) and +4.31 with nonexpert. This proves the expert's quality is vital.

---

> ### Author Response · Authors · 2025-11-20
> **Response to Reviewer i1wX (part2)**
>
> > **Q6:** How would the framework scale to models with hundreds of layers or significantly higher hidden dimensions?
> > >
> > > **R6:**  In our R2 response, we employed Llama-3.3-70B-Instruct as the target model, which has 80 layers and a hidden dimension of 8192. Our medical expert model has 32 layers and a hidden dimension of 4096. The results confirm that ExpertSteer remains effective and computationally feasible even at this scale. Due to computational resource constraints, Llama-3.3-70B-Instruct is the largest model we could currently test. We believe this is sufficient to demonstrate the scalability of our method to very large models. Furthermore, to the best of our knowledge, there is no mainstream open-source LLM with more than 100 layers. For instance, Qwen/Qwen3-235B-A22B-Instruct-2507 has only 94 layers, openai/gpt-oss-120b has only 36 layers, moonshotai/Kimi-K2-Thinking has only 61 layers, and mistralai/Mistral-Large-Instruct-2411 has only 88 layers.
>
> > **Q7:** Can the approach be adapted for multimodal models or instruction-following alignment tasks?
> > >
> > > **R7:**  For adapting to the multimodal models, it is out of scope of this paper. While we focused on text, our framework is general. As our ExpertSteer only intervenes the text generation process at the inference, we believe it can be easily extended to multimodal models. For example, it could align the text representation spaces of a target model and a superior multimodal expert to improve text-based reasoning in a multimodal context.
> > >
> > > To evaluate the instruction-following ability, we ran a new supplementary experiment using Qwen2.5-14B-Instruct as an expert and Llama-3.1-8b-Instruct as the target on the IFEval benchmark:
> > >
> |   model    | accuracy |
> |---|---|
> | baseline  |    80.4    |
> |     ExpertSteer (with general expert)         |    82.2    |
> > >
> > > This confirms that ExpertSteer is a highly effective and low-cost method for alignment tasks.
>
> > **Q8:** Does applying external steering vectors risk transferring unintended biases from the expert model?
> > >
> > > **R8:**  Please see our response R4 to Q4.

---

### Comment · Area_Chair_MPmv · 2025-11-23
**Next Steps Following Authors’ Rebuttal: Review Rebuttal and Participate in Discussion**

Dear Reviewers,

Thank you very much for your thoughtful evaluations of this paper.

Now that the authors have submitted their rebuttal, I kindly ask you to take the following steps (if you have not done so already):

- Read the other reviews as well as the authors’ response.
- Consider whether the rebuttal and additional comments affect your assessment of the paper.
- Engage in interactive discussion with the authors **before November 25**, encouraging a dynamic exchange rather than a one-sided rebuttal.

The current reviews for this paper are mixed. Your contributions at this stage are essential for forming a well-informed final decision. I therefore ask that you reassess your views in light of the authors’ responses and the broader discussion among reviewers.

I am happy to join and support the discussions between you and the authors. Please feel free to share your thoughts and participate actively in the discussion.

Thank you once again for your service to ICLR 2026.

Best regards,

 AC

---

### Meta-Review · Area_Chair_EC2U · 2026-01-03

**Summary:**

This paper proposes ExpertSteer, which transfers steering vectors from external expert LLMs to target LLMs via alignment, MI layer pairing, RFM extraction at inference.

**Reviewer Concerns:**

Main concerns from reviewers include:

1. MI-based layer pairing and RFM choices feel ad hoc / weakly justified;

2. Limited mechanistic/causal analysis; dependence on expert quality and sizable labeled data;

3. Hyperparameter search and per-pair adapters weaken “lightweight/model-agnostic” claims;

4. Missing stronger PEFT baselines (e.g., LoRA); limited large-model/robustness evaluation.

**Reviewer Scores:**

Reviewer / Score

i1wX	6

1afh	4

46Sn	4

14cR	2

oDcb	4

Average	4

No reviewers indicated to increase or decrease their scores.

---

### Decision · Program_Chairs · 2026-01-26

Reject